# Robust agents learn causal world models

**Jonathan Richens**[*]
Google DeepMind

**Tom Everitt**
Google DeepMind

## Abstract

It has long been hypothesised that causal reasoning plays a fundamental role in robust and general intelligence. However, it is not known if agents must learn causal models in order to generalise to new domains, or if other inductive biases are sufficient. We answer this question, showing that any agent capable of satisfying a regret bound for a large set of distributional shifts must have learned an approximate causal model of the data generating process, which converges to the true causal model for optimal agents. We discuss the implications of this result for several research areas including transfer learning and causal inference.

## 1 Introduction

What capabilities are necessary for general intelligence (Legg & Hutter, 2007)? One candidate is causal reasoning, which plays a foundational role in human cognition (Gopnik et al., 2007; Sloman & Lagnado, 2015). It has even been argued that human-level AI is impossible without causal reasoning (Pearl, 2018). However, recent years have seen the development of agents that do not explicitly learn or reason on causal models, but nonetheless are capable of adapting to a wide range of environments and tasks (Reed et al., 2022; Team et al., 2023; Brown et al., 2020).

This raises the question, do agents have to learn causal models in order to adapt to new domains, or are other inductive biases sufficient? To answer this question, we have to be careful not to assume that agents use causal assumptions a priori. For example, transportability theory determines what causal knowledge is necessary for transfer learning when all assumptions on the data generating process (inductive biases) can be expressed as constraints on causal structure (Bareinboim & Pearl, 2016). However, deep learning algorithms can exploit a much larger set of inductive biases (Neyshabur et al., 2014; Battaglia et al., 2018; Rahaman et al., 2019; Goyal & Bengio, 2022) which in many real-world tasks may be sufficient to identify low regret policies without requiring causal knowledge.

The main result of this paper is to answer this question by showing that,

> *Any agent capable of adapting to a sufficiently large set of distributional shifts must have learned a causal model of the data generating process.*

Here, adapting to a distributional shift means learning a policy that satisfies a regret bound following an intervention on the data generating process—for example, changing the distribution of features or latent variables. It is known that a causal model of the data generating process can be used to identify regret-bounded policies following a distributional shift (sufficiency), with more accurate models allowing lower regret policies to be found. We prove the converse (necessity)—given regret-bounded policies for a large set of distributional shifts, we can learn an approximate causal model of the data generating process, with the approximation becoming exact for optimal policies. Hence, learning a causal model of the data generating process is necessary for robust adaptation.

This has consequences for a number of fields and questions. For one, it implies that causal identification laws also constrain domain adaptation. For example, we show that adapting to covariate and label shifts is only possible if the causal relations between features and labels can be identified from the training data—a non-trivial causal discovery problem. This provides further theoretical justification for causal representation learning (Schölkopf et al., 2021), showing that learning causal representations is necessary for achieving strong robustness guarantees. Our result also implies that we can learn causal models from adaptive agents. We demonstrate this by solving a causal discovery

---

[*]jonrichens@deepmind.com

task on synthetic data by observing the policy of a regret-bounded agent under distributional shifts. More speculatively, our results suggest that causal models could play a role in emergent capabilities. Agents trained to minimise a loss function across many domains are incentivized to learn a causal world model, which could enable them to solve a much larger set of decision tasks they were not explicitly trained on.

**Outline of paper.** In Section 2 we introduce concepts from causality and decision theory used to derive our results. We present our main theoretical results in Section 3 and discuss their interpretation in terms of adaptive agents, transfer learning and causal inference. In Section 4 we discuss limitations, as well as implications for a number of fields and open questions. In section 5 we discuss related work including transportability (Bareinboim & Pearl, 2016) and the causal hierarchy theorem (Bareinboim et al., 2022), and recent empirical work on emergent world models. In Appendix B we describe experiments applying our theoretical results to causal discovery problems.

## 2 PRELIMINARIES

### 2.1 CAUSAL MODELS

We use capital letters for random variables $V$, and lower case for their values $v \in dom(V)$. For simplicity, we assume each variable has a finite number of possible values, $|dom(V)| < \infty$. Bold face denotes sets of variables $\mathbf{V} = \{V_1, \ldots, V_n\}$, and their values $\mathbf{v} \in dom(\mathbf{V}) = \times_i dom(V_i)$. A probabilistic model specifies the joint distribution $P(\mathbf{V})$ over a set of variables $\mathbf{V}$. These models can support associative queries, for example $P(\mathbf{Y} = \mathbf{y} \mid \mathbf{X} = \mathbf{x})$ for $\mathbf{X}, \mathbf{Y} \subseteq \mathbf{V}$. Interventions describe external changes to the data generating process (and hence changing the joint distribution), for example a *hard* intervention $do(\mathbf{X} = \mathbf{x})$ describes forcing the set of variables $\mathbf{X} \subseteq \mathbf{V}$ to take value $\mathbf{x}$. This generates a new distribution $P(\mathbf{V} \mid do(X = x)) = P(\mathbf{V}_x)$ where $\mathbf{V}_{\mathbf{x}}$ refers to the variables $\mathbf{V}$ following this intervention. The power of causal models is that they specify not only $P(\mathbf{V})$ but also the distribution of $\mathbf{V}$ under all interventions, and hence these models can be used to evaluate both associative and interventional queries e.g. $P(\mathbf{Y} = \mathbf{y} \mid do(\mathbf{X} = \mathbf{x}))$.

For the derivation of our results we focus on a specific class of causal models—causal Bayesian networks (CBNs). There are several alternative models and formalisms that are studied in the literature, including structural equation models (Pearl, 2009) and the Neyman-Rubin causal models (Rubin, 2005), and results can be straightforwardly adapted to these.

**Definition 1** (Bayesian networks). *A Bayesian network $M = (G, P)$ over a set of variables $\mathbf{V} = \{V_1, \ldots, V_n\}$ is a joint probability distribution $P(\mathbf{V})$ that factors according to a directed acyclic graph (DAG) $G$, i.e. $P(V_1, \ldots, V_n) = \prod_{i=1}^{n} P(V_i \mid \mathbf{Pa}_{V_i})$, where $\mathbf{Pa}_{V_i}$ are the parents of $V_i$ in $G$.*

A Bayesian network is *causal* if the graph $G$ captures the causal relationships between the variables or, formally, if the result of any intervention $do(\mathbf{X} = \mathbf{x})$ for $\mathbf{X} \subseteq \mathbf{V}$ can be computed from the truncated factorisation formula:

$$P(\mathbf{v} \mid do(\mathbf{x})) = \begin{cases} \prod_{i : v_i \notin \mathbf{x}} P(v_i \mid \mathbf{pa}_{v_i}) & \text{if } \mathbf{v} \text{ consistent with } \mathbf{x} \\ 0 & \text{otherwise.} \end{cases}$$

More generally, a *soft* intervention $\sigma_{v_i} = P'(V_i \mid \mathbf{Pa}_i^*)$ replaces the conditional probability distribution for $V_i$ with a new distribution $P'(V_i \mid \mathbf{Pa}_i^*)$, possibly resulting in a new parent set $\mathbf{Pa}_i^* \neq \mathbf{Pa}_i$ as long as no cycles are introduced in the graph. We refer to $\sigma_{v_i}$ as a *domain indicator* (Correa & Bareinboim, 2020) (it has also been called an environment index, Arjovsky et al., 2019). The updated distribution is denoted $P(\mathbf{v}; \sigma_{\mathbf{v}'}) = \prod_{i : v_i \in \mathbf{v}'} P'(v_i \mid \mathbf{pa}_{v_i}^*) \prod_{i : v_i \notin \mathbf{v}'} P(v_i \mid \mathbf{pa}_{v_i})$.

In general, soft interventions cannot be defined without knowledge of $G$. For example, the soft intervention $\sigma_Y = P'(y \mid x)$ is incompatible with the causal structure $Y \to X$ as it would induce a causal cycle. As our results are concerned with learning causal models (and hence causal structure), we focus our theoretical analysis on a subset of the soft interventions, *local interventions*, that are compatible with all causal structures and so can be used without tacitly assuming knowledge of $G$.

**Definition 2** (Local interventions). *Local intervention $\sigma$ on $V_i \in \boldsymbol{V}$ involves applying a map to the states of $V_i$ that is not conditional on any other endogenous variables, $v_i \mapsto f(v_i)$. We use the notation $\sigma = do(V_i = f(v_i))$ (variable $V_i$ is assigned the state $f(v_i)$). Formally, this is a soft intervention on $V_i$ that transforms the conditional probability distribution as,*

$$P(v_i \,|\, \boldsymbol{pa}_i; \sigma) = \sum_{v_i':f(v_i')=v_i} P(v_i' \,|\, \boldsymbol{pa}_i) \tag{1}$$

*Example:* Hard interventions $do(V_i = v_i')$ are local interventions where $f(v_i)$ is a constant function.

*Example:* Translations are local interventions as $do(V_i = v_i + k) = do(V_i = f(v_i))$ where $f(v_i) = v_i + k$. Examples include changing the position of objects in RL environments (Shah et al., 2022) and images (Engstrom et al., 2019).

*Example:* Logical NOT operation $X \mapsto \neg X$ for Boolean $X$ is a local intervention.

We also consider stochastic interventions, noting that mixtures of local interventions can also be defined without knowledge of $G$. For example, adding noise to a variable $X = X + \epsilon, \epsilon \sim \mathcal{N}(0,1)$, is a soft intervention on $X$ described by a mixture over local interventions (translations).

**Definition 3** (Mixtures of interventions). *A mixed intervention $\sigma^* = \sum_i p_i \sigma_i$ for $\sum p_i = 1$ performs intervention $\sigma_i$ with probability $p_i$. Formally, $P(\boldsymbol{v} \,|\, \sigma^*) = \sum_i p_i P(\boldsymbol{v} \,|\, \sigma_i)$.*

## 2.2 Decision tasks

Decision tasks involve a decision maker (agent) choosing a policy so as to optimise an objective function (utility). To give a causal description of decision tasks we use the causal influence diagram (CID) formalism (Howard & Matheson, 2005; Everitt et al., 2021), which extend a CBN of the environment (chance) variables by introducing decision and utility nodes (see Figure 1 for examples). For simplicity we focus on tasks involving a single decision and a single utility function.

**Definition 4** (Causal influence diagram). *A (single-decision, single-utility) causal influence diagram (CID) is a CBN $M = (G, P)$ where the variables $\boldsymbol{V}$ are partitioned into decision, utility, and chance variables, $\boldsymbol{V} = (\{D\}, \{U\}, \boldsymbol{C})$. The utility variable is a real-valued function of its parents, $U(\boldsymbol{pa}_U)$.*

Single-decision single-utility CIDs can represent most decision tasks such as classification and regression as they specify what decision should be made ($d \in D$), based on what information ($\mathbf{pa}_D$), with objective ($\mathbb{E}[U]$). They can also describe some multi-decision tasks such as Markov decision processes[1]. The utility is any real-valued function including standard loss and reward functions.

We assume that the environment is described by a set of random variables $\boldsymbol{C}$ that interact via causal mechanisms[2], and where $\boldsymbol{C}$ satisfies causal sufficiency (Pearl, 2009) (includes all common causes), noting that such a choice of $\boldsymbol{C}$ always exists. We refer to the CBN over $\boldsymbol{C}$ as the 'true' or 'underlying' CBN. Note we do not assume the agent has any knowledge of the underling CBN, nor do we assume which variables in $\boldsymbol{C}$ are observed or unobserved by the agent, beyond that the agent can observe $\mathbf{Pa}_D \subseteq \boldsymbol{C}$. We also assume knowledge of the utility function $U(\mathbf{Pa}_U)$.

The conditional probability distribution for the decision node $\pi(d \,|\, \mathbf{pa}_D)$ (the policy) is not a fixed parameter of the model but is set by the agent so as to maximise its expected utility, which for a policy $\pi$ is $\mathbb{E}^\pi[U] = \mathbb{E}[U \,|\, do(D = \pi(\mathbf{pa}_D))]$. A policy $\pi^*$ is *optimal* if it maximises $\mathbb{E}^{\pi^*}[U]$. Typically, agents do not behave optimally and incur some *regret* $\delta$, which is the decrease in expected utility compared to an optimal policy $\delta := \mathbb{E}^{\pi^*}[U] - \mathbb{E}^\pi[U]$.

To simplify our theoretical analysis, we focus on a widely studied class of decision tasks where the agents decision does not causally influence the environment (e.g. Figure 1).

**Assumption 1** (Unmediated decision task). $\boldsymbol{Desc}_D \cap \boldsymbol{Anc}_U = \emptyset$.

In unmediated decision tasks, the agent is provided some (partial) observations of the environment and chooses a policy, which is then evaluated using the utility function which is a function of the

---

[1]Note Markov decision processes can be formulated as a single-decision single-utility CID, by modelling the choice of policy as a single decision and the cumulative discounted reward as a single utility variable.

[2]This assumption follows from Reichenbach (1956), and we discuss further in Appendix A.3

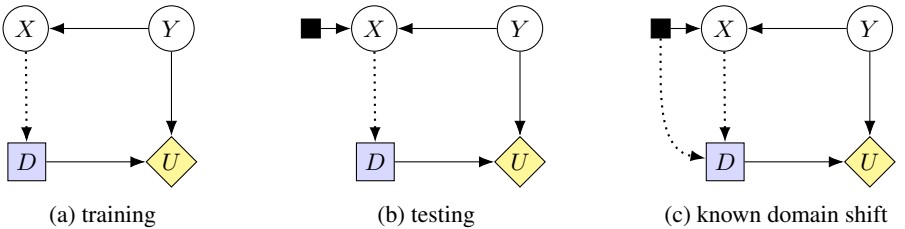

(a) training  (b) testing  (c) known domain shift

Figure 1: CID for a supervised learning task during (a) training and (b) testing following a distributional (covariate) shift (unsupervised domain adaptation, Wilson & Cook, 2020). The agent chooses a label prediction $D = \hat{Y}$ given features $X$, with the goal of minimising loss $U = -\text{Loss}(Y, \hat{Y})$. Decision variables are depicted as square nodes, chance variables as circular nodes and utilities as diamond nodes. Information edges (dashed) show the variables the agent conditions their policy on. In this example the labels cause the features $Y \rightarrow X$ (for examples where features cause labels see Castro et al., 2020; Schölkopf et al., 2012). The black square ('regime node' (Correa & Bareinboim, 2020)) in (b) and (c) denotes a distributional shift induced by an intervention on $X$. Diagram (c) depicts the idealised case where the agent knows what domain shift has occurred. By theorem 1, if the agent can return an optimal decision boundary for known covariate and label shifts, then it must have learned the CBN over $\boldsymbol{C} = \{X, Y\}$. Note that even if the agent has sufficient training data to learn $P(X, Y)$, the causal structure $Y \rightarrow X$ is in general non-identifiable given $P(X, Y)$ and so domain adaptation requires that the agent solves a non-trivial causal discovery problem.

environment state and the agent's decision. Examples of unmediated decision tasks include prediction tasks such as classification and regression, whereas examples of *mediated* decision tasks that are not covered by our theorems include Markov decision processes where the agent's decision (action) influences the utility via the environment state.

## 2.3 DISTRIBUTIONAL SHIFTS

We focus on generalisation that goes beyond the *iid* assumption, where agents are evaluated in domains that are distributionally shifted from the training environment. Distributional shifts can be changes to the environment (*domain shifts*), as in domain adaptation and domain generalisation (Farahani et al., 2021; Wilson & Cook, 2020), or changes to the objective (*task shifts)* as in zero shot learning (Xian et al., 2018), in-context learning (Brown et al., 2020) and multi-task reinforcement learning (Reed et al., 2022). Our analysis focuses on domain shifts that involve changes to the causal data generating process, and hence can be modelled as interventions (Schölkopf et al., 2021). This does not assume that all shifts an agent will encounter can be modelled as interventions, but requires that the agent is *at least* capable of adapting to these shifts.

Examples of interventionally generated shifts include translating objects in images (Engstrom et al., 2019), noising inputs and adversarial robustness (Hendrycks & Dietterich, 2019), and changes to the initial conditions or transition function in Markov decision processes (Peng et al., 2018). Examples of shifts that are not naturally represented as interventions include changing the set of environment variables $\boldsymbol{C}$, and introducing selection biases (Shen et al., 2018). See Appendix A.3 for discussion.

Our main results restrict to local *domain shifts*, which correspond to local interventions on the chance variables $\boldsymbol{C}$. We do not consider shifts that change the agent's decision $D$, although we include shifts that drop inputs to the policy $\mathbf{Pa}_D \rightarrow \mathbf{Pa}'_D \subseteq \mathbf{Pa}_D$ (e.g. masking) as local interventions. We do not consider task shifts i.e. changing the utility function.

As we are interested in determining the capabilities necessary for domain adaptation, we restrict our attention to decision tasks where domain adaptation is non-trivial, i.e. where the optimal policy depends on the environment distribution $P(\boldsymbol{C} = \boldsymbol{c})$.

**Assumption 2** (Domain dependence). *There exists $P(\boldsymbol{C} = \boldsymbol{c})$ and $P'(\boldsymbol{C} = \boldsymbol{c})$ compatible with $M$ such that $\pi^* = \arg\max_\pi \mathbb{E}_P^\pi[U]$ implies $\pi^* \neq \arg\max_\pi \mathbb{E}_{P'}^\pi[U]$.*

Assumption 2 implies the existence of domain shifts that change the optimal policy.

## 3 CAUSAL MODELS ARE NECESSARY FOR ROBUST ADAPTATION

We now present our results, showing that learning the underlying CBN is necessary and sufficient for learning regret bounded policies under a large set of domain shifts. In Section 3.2 we interpret these results by considering their consequences for transfer learning, causal inference and adaptive agents.

First we focus on the idealised case where we assume optimality. We show for almost all decision tasks the underlying CBN can be reconstructed given optimal policies for a large set of domain shifts.

**Theorem 1.** *For almost all CIDs $M = (G, P)$ satisfying Assumptions 1 and 2, we can identify the directed acyclic graph $G$ and joint distribution $P$ over all ancestors of the utility $\mathbf{Anc}_U$ given $\{\pi_\sigma^*(d \mid \mathbf{pa}_D)\}_{\sigma \in \Sigma}$ where $\pi_\sigma^*(d \mid \mathbf{pa}_D)$ is an optimal policy in the domain $\sigma$ and $\Sigma$ is the set of all mixtures of local interventions. Proof in Appendix C.*

The parameters $P(v_i \mid \mathbf{pa}_i)$, $U(\mathbf{pa}_U)$ of the underlying CBN define a parameter space and the condition *for almost all CIDs* means that the subset of the parameter space for which the Theorem 1 does not hold is Lebesgue measure zero (see Appendix A.2 for discussion). This condition is necessary because there exist finely-tuned environments for which the CBN cannot be identified given the agent's policy due to variables $X \in \mathbf{Anc}_U$ that do not affect the expected utility. For example consider $X \to Y \to U$, $Y = \mathcal{N}(0, x)$ and $U = D + Y$, then changing $X$ can only change the variance of $U$ while leaving its expected value (and hence the optimal policy) constant. However, this only occurs for very specific choices of the parameters $P$ and $U$.

In Appendix B we give a simplified overview of the proof with a worked example. We assume access to an oracle for optimal policies $\pi_\sigma^*$ for any given local intervention on $\mathbf{C}$. Note, this assumes the agent is robust to distributional shifts on a causally sufficient set of variables $\mathbf{C}$, not that the set of variables the agent observes is causally sufficient. We devise an algorithm that queries this oracle with different mixtures of local interventions and identifies the mixtures for which the optimal policies changes. We then show that these critical mixtures identify the parameters of the CBN, specifying both the graph $G(\mathbf{Anc}_U)$ and the joint distribution $P(\mathbf{Anc}_U)$.

### 3.1 RELAXING THE ASSUMPTION OF OPTIMALITY

We now relax the assumption of optimality, considering the case where the policies $\pi_\sigma$ satisfy a regret bound $\mathbb{E}^{\pi_\sigma}[U] \geq \mathbb{E}^{\pi_\sigma^*}[U] - \delta$. We show that for $\delta > 0$ we can recover an approximation of the environment CBN, with error that grows linearly in $\delta$ for $\delta \ll \mathbb{E}^{\pi^*}[U]$.

**Theorem 2.** *For almost all CIDs $M = (G, P)$ satisfying Assumptions 1 and 2, we can identify an approximate causal model $M' = (P', G')$ given $\{\pi_\sigma(d \mid \mathbf{pa}_D)\}_{\sigma \in \Sigma}$ where $\mathbb{E}^{\pi_\sigma}[U] \geq \mathbb{E}^{\pi_\sigma^*}[U] - \delta$ and $\Sigma$ is the set of mixtures of local interventions. The parameters of $M'$ satisfy $|P'(v_i \mid \mathbf{pa}_i) - P(v_i \mid \mathbf{pa}_i)| \leq \gamma(\delta) \; \forall \; V_i \in \mathbf{V}$ where $\gamma(0) = 0$ and $\gamma(\delta)$ grows linearly in $\delta$ for small regret $\delta \ll \mathbb{E}^{\pi^*}[U]$. Proof in Appendix D.*

The worst-case error bounds $\gamma(\delta)$ for the parameter errors are detailed in Appendix D. For $\delta > 0$ it may not be possible to identify $G$ perfectly as some weak causal relations cannot be resolved due to these error bounds. We describe in Appendix D how we can learn a sub-graph $G' \subseteq G$ that may exclude directed edges corresponding to weak causal relations.

Theorem 2 shows that we can learn a (sparse) approximate causal models of the data generating process from regret bounded policies under domain shifts, where the approximation becoming exact as $\delta \to 0$. In Appendix F we demonstrate learning the underlying CBN from regret-bounded policies using simulated data for randomly generated CIDs similar to Figure 1, and explore how the accuracy of the approximate CBN scales with the regret bound (Figure 3).

Finally, we prove sufficiency, i.e. that having an (approximate) causal model of the data generating process is sufficient to identify regret-bounded policies. The result is well-known for the non-approximate case (Bareinboim & Pearl, 2016).

**Theorem 3.** *Given the CBN $M = (P, G)$ that is causally sufficient we can identify optimal policies $\pi_\sigma^*(d \mid \mathbf{pa}_D)$ for any given $U$ where $\mathbf{Pa}_U \subseteq \mathbf{C}$ and for all soft interventions $\sigma$. Given an approximate causal model $M' = (P', G')$ for which $|P'(v_i \mid \mathbf{pa}_i) - P(v_i \mid \mathbf{pa}_i)| \leq \epsilon \ll 1$, we can identify regret-bounded policies where the regret $\delta$ grows linearly in $\epsilon$. Proof in Appendix E.*

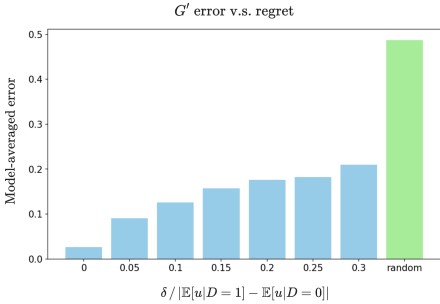
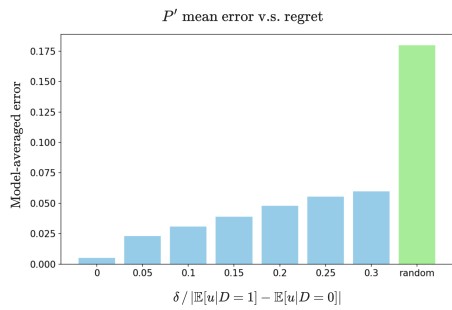

(a) Error rate for learned DAG v.s. regret bound

(b) Mean error for $P(x, y)$ v.s. regret bound

Figure 3: Comparing the model-average error rates for a) the learned DAG $G'$ and b) learned joint distribution $P'(x, y)$, v.s. the (normalised) regret bound $\delta / \big| \mathbb{E}[u \mid D = 1] - \mathbb{E}[u \mid D = 0] \big|$. Average error taken over 1000 randomly generated environments with binary decision $D$ and two binary latent variables $X, Y$. Comparison to error rate for random guess (green) See Appendix F for details.

Together, Theorems 2 and 3 imply that learning an approximate causal model of the data generating process is necessary and sufficient for learning regret-bounded policies under local interventions.

## 3.2 INTERPRETATION

We interpret Theorems 1 to 3 through three lenses; agents, transfer learning and causal inference.

**Agents** are adaptive goal-directed systems (Dennett, 1989), meaning they choose actions they expect to achieve some desired outcome, and would adapt their behaviour if they knew the consequences of their actions had changed. For example, a firm sets prices to maximise profit, and adapts pricing to changes to supply and demand (Kenton et al., 2023).

Consider an agent capable of zero-shot generalisation under domain shifts (Kirk et al., 2023). First, consider the case where the agent has perfect knowledge of the domain shift, i.e. the policy conditions on the domain indicator[3] $\pi_\sigma = \pi(d \mid \mathbf{pa}_D, \sigma)$. By Theorem 2 if $\pi_\sigma$ satisfies a tight regret bound for all local shifts $\sigma$, we can reconstruct the underlying CBN from the agent's policy alone (following the procedure in Appendix C). Therefore, any agent capable of generalising under known local domain shifts must have learned the underling CBN. Precisely, the agent has learned the policy $\pi(d \mid \mathbf{pa}_D, \sigma)$, which is functionally equivalent to learning the underlying CBN, as any associative or causal query that can be identified using the CBN can also be identified using $\pi(d \mid \mathbf{pa}_D, \sigma)$. If $\sigma$ is unknown and has to be inferred by the agent, the underlying CBN can similarly reconstructed from the agent's policy as long as it satisfies a tight regret bound[4].

*Example:* Doctors are agents expected to make low regret decisions under a wide range of known distributional shifts, without re-training in the shifted environment. For example, consider the task of risk-stratifying patients based on their signs and medical history. The doctor may be transferred to a new ward where patients have received a treatment (known distributional shift) that has a stochastic effect on latent variables (mixed intervention) such as curing diseases and causing side effects. The doctor cannot re-train in this new domain, e.g. taking random decisions and observing outcomes. To be capable of this, Theorem 2 implies the doctor must have learned the causal relations between the relevant latent variables—how the treatment affects diseases, how these diseases and their symptoms are causally related, and so on. Likewise, any medical AI that hopes to replicate this capability must have learned a similarly accurate causal model, and the better the agent's performance the more accurate its causal model must be.

---

[3]equivalently an environment index for (in-context) invariant risk minimization (Gupta et al., 2023; Arjovsky et al., 2019), or the context variable for zero-shot reinforcement learning (Kirk et al., 2023)

[4]Trivially, any agent capable of returning a regret-bounded policy without input $\sigma$ can do so given $\sigma$ simply by discarding $\sigma$. If the agent adapts by conditioning their policy on an inferred $\sigma'$ (see Kirk et al. (2023) for a review of implementations), and this policy satisfies a regret bound, it can be used in Theorems 1 and 2.

**Transfer learning.** In transfer learning (Zhuang et al., 2020), models are trained on a set of source domains and evaluated on held-out target domains where i) the data distribution differs from the source domains, and ii) the data available for training is restricted compared to the source domains (Wang et al., 2022). For example, in unsupervised domain adaptation the learner is restricted to samples of the input features from the target domains $\mathbf{pa}_D \sim P(\mathbf{Pa}_D; \sigma)$, whereas in domain generalisation typically no data from the target domain is available during training (Farahani et al., 2021).

Let $\mathcal{D}_S$ denote the training data from the source domains and $\mathcal{D}_\sigma$ denote the training data available from a given target domain $\sigma$. Let there exist a transfer learning algorithm that returns a policy $\pi_\sigma$ satisfying a regret bound for a given target domain $\sigma$, provided this training data. As $\pi_\sigma$ is a function of the training data, then by Theorems 1 and 2 the existence of this algorithm implies that we can identify the underlying CBN from $\mathcal{D}_S \cup \{\mathcal{D}_\sigma\}_{\sigma \in \Sigma}$. To see that this imparts non-trivial constraints on the existence of the transfer learning algorithm, we can consider the following simple example.

*Example:* Consider the CID for the supervised learning task depicted in Figure 1. Let $\mathcal{D}_S = \{(x^i, y^i) \sim P(X, Y)\}_{i=1}^n$, so for sufficiently large $n$ the agent can learn the $P(X, Y)$ from $\mathcal{D}_S$. However, $Y \to X$ must also be identifiable from the training data $\mathcal{D}_S \cup \{\mathcal{D}_\sigma\}_{\sigma \in \Sigma}$. In other words, the transfer learning problem contains a hidden causal discovery problem. If $\mathcal{D}_\sigma = \emptyset$ then $Y \to X$ must be identifiable from $P(x, y)$ alone, which is impossible unless the causal data generating process obeys additional assumptions (see for example Hoyer et al., 2008). If unlabelled features from the target domain are included in the training data $\mathcal{D}_\sigma = \{x^i \sim P(X; \sigma)\}_{i=1}^{n_\sigma}$, $Y \to X$ can in principle be identified as $P(X; \sigma_Y) \neq P(X)$.

**Causal inference.** Theorem 1 can also be interpreted purely in terms of causal inference. We can compare to the causal hierarchy theorem (CHT) (Bareinboim et al., 2022), which states that an oracle for L1 queries (observational) is almost always insufficient to evaluate all L2 queries (interventional). Our Theorem 1 can be stated in an analogous way; an oracle for optimal policies under mixtures of local interventions $\Pi_\Sigma^* : \sigma \mapsto \pi^*(\sigma)$, can evaluate all L2 queries, which follows from the fact that the oracle identifies the underlying CBN which in turn identifies all L2 queries. Note $\Pi_\Sigma^*$ is a strict subset of L2, and we describe a subset of L2 as being *L2-complete* if evaluating these queries is sufficient to evaluate all L2 queries. Hence Theorem 1 can be summarised as $\Pi_\Sigma^*$ is L2-complete. It would be interesting in future work to determine what other strict subsets of L2 are L2-complete, as identifying these queries is sufficient to identify all interventional queries.

Why is this surprising? Firstly, we may expect the optimal policies to encode a relatively small number of causal relations, as they can be computed from $\mathbb{E}[u \mid d, \mathbf{pa}_D; \sigma]$, which describes the response of a single variable $U$ to intervention $\sigma$. However, Theorem 1 shows that the optimal policies encode all causal and associative relations in $\mathbf{Anc}_U$, including causal relations between latent variables, for example $P(\mathbf{Y}_x)$ for any $\mathbf{X}, \mathbf{Y} \subseteq \mathbf{Anc}_U$. Secondly, Theorems 2 and 3 combined imply that learning to generalise under domain shifts is *equivalent* to learning a causal model of the data generating process—problems that on the surface are conceptually distinct.

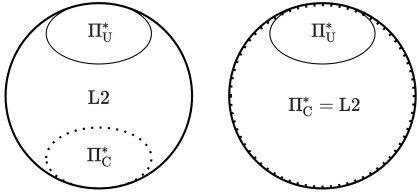

Figure 4: The set L2 (Bareinboim et al., 2022) contains all casual queries including the set of optimal policies under domain $\boldsymbol{\pi}_C^*$ and task shifts $\boldsymbol{\pi}_U^*$. We show that $\boldsymbol{\pi}_U^*$ also contains L2, i.e. learning optimal policies under all shifts for a single utility $U$ is sufficient to identify L2

## 4    DISCUSSION

Here we discuss the consequences for several fields and open questions, as well as limitations.

**Causal representation learning.** Causal representation learning (CRL) aims to learn representations of data that capture unknown causal structure (Schölkopf et al., 2021), with the aim of exploiting causal invariances to achieve better generalisation across domains. Theorems 1 and 2 show that any method that enables generalisation across many domains necessarily involves learning an (approximate) causal model of the data generating process—i.e. a causal representation. Hence, our results provide theoretical justification for CRL by showing it is necessary for strong robustness guarantees.

**Causal bounds on transfer learning.** As described in Section 3.2, Theorems 1 and 2 imply fundamental causal constraints on certain transfer learning tasks. For example in the supervised learning task depicted in Figure 1, identifying regret-bounded policies under covariate and label shifts requires learning the causal relations between features and labels. Causal discovery problems such as this are well understood in many settings (Vowels et al., 2022), and in general identifying this causal structure (e.g. that $Y \rightarrow X$ in Figure 5 a)) is impossible without interventional data and/or additional assumptions. This connection allows us to convert (im)possibility results for causal discovery to (im)possibility results for transfer learning. Future work could explore this for smaller sets of distributional shifts and derive more general causal bounds on transfer learning.

**Good regulator theorem.** The good regulator theorem is often interpreted as saying that any good controller of a system must have a model of that system (Conant & Ross Ashby, 1970). However, some imagination is needed to take this lesson from the actual theorem, which technically only states that there exists an optimal regulator that is a deterministic function of the state of the system (which could be trivial, Wentworth (2021)). Our theorem less ambiguously states that any robust agent must have learned an (approximate) causal model of the environment, as described in Section 3.2. It can therefore be interpreted as a more precise, causal good regulator theorem.

**Emergent capabilities.** Causal models enable a kind of general competency—an agent can use a causal model of its environment to optimise for any given objective function $U(\mathbf{Pa}_U \subseteq \mathbf{V})$ without additional data (Theorem 3). This could explain how general competence can arise from narrow training objectives (Brown et al., 2020; Silver et al., 2021). By Theorems 1 and 2, agents trained to maximise reward across many environments are incentivized to learn a causal world model (as they cannot generalise without one), which can in turn be used to solve any other decision task in the same environment. This incentive does not imply that training an agent with a simple reward signal is sufficient to learn causal world models. E.g. it will still be impossible for an agent to learn a causal model (and therefore to generalise) if the model is not identifiable from its training data. The question is then if current methods and training schemes are sufficient for learning causal world models. Early results suggest that transformer models can learn world models capable of out-of-distribution prediction (Li et al., 2022, see Section 6 for discussion). While foundation models are capable of achieving state of the art accuracy on causal reasoning benchmarks (Kıcıman et al., 2023), how they achieve this (and if it constitutes bona fide causal reasoning) is debated (Zečević et al., 2023).

**Causal discovery.** Theorems 1 and 2 involve learning the causal structure of the environment by observing the agent's policy under interventions. It is perhaps surprising that the response of this single variable to interventions is sufficient to identify all associative and causal relations in $\mathbf{Anc}_U$. Typically, causal discovery algorithms involve measuring the response of many variables to interventions (Vowels et al., 2022). Also, many causal discovery algorithms assume independent causal mechanisms (Schölkopf et al., 2021), which is equivalent to assuming no agents are present in the data generating process (Kenton et al., 2023). However, our results suggest that agents could be powerful resources for causal discovery. In Appendix B we use the proof of Theorem 2 to derive a causal discovery algorithm for learning causal structure over latents, and test it on synthetic data.

**Applicability of causal methods.** Causal models have been used to formally define concepts such as intent (Halpern & Kleiman-Weiner, 2018; Ward et al., 2024), harm (Richens et al., 2022), deception (Ward et al., 2023b), manipulation (Ward et al., 2023a) and incentives (Everitt et al., 2021), and are required for approaches to explainability (Wachter et al., 2017) and fairness (Kusner et al., 2017). Methods for designing safe and ethical AI systems that build on these definitions require causal models of the data generating process, which are typically hard to learn, leading some to doubt their practicality (Fawkes et al., 2022; Rahmattalabi & Xiang, 2022). However, our results show that learning a causal model of the data generating process is necessary for training robust agents, and

demonstrate that these models can be elicited from these agents. These models could in turn be used to support causal methods for aligning agents.

**Limitations.** Theorems 1 and 2 require agents to be robust to a large set of domain shifts (local interventions on all environment variables). Theorem 2 shows that loosening regret bounds results in some causal relations being unidentifiable from the agent's policy. Hence, we expect it is still possible to learn some casual knowledge of the environment from agents that are robust to a smaller set of domain shifts, albeit less complete that the full underlying CBN. Finally, our results only apply to unmediated decision tasks (Assumption 1). We expect Theorems 1 and 2 can be extended to active decision tasks, as Assumption 1 does not play a major role beyond simplifying the proofs.

## 5 RELATED WORK

Several recent empirical works have explored if deep learning models learn 'surface statistics' (e.g. correlations between inputs and outputs) or learn internal representations of the world (McGrath et al., 2022; Abdou et al., 2021; Li et al., 2022; Gurnee & Tegmark, 2023). Our results offer some theoretical clarity to this discussion, tying an agents performance to the fidelity its world model, and that going beyond 'surface statistics' to learning causal relations is fundamentally necessary for robustness. One study in particular (Li et al., 2022) found that a GPT model trained to predict legal next moves in the board game Othello learned a linear representation of the board state (Nanda, 2023). Further, this internal representation of the board state could be changed by intervening on the intermediate activations, with the model updating its predictions consistent with the intervention, including interventions that take the board state outside of the training distribution. This indicates that the network is learning and utilising a representation of the data generating process that can support out-of-distribution generalisation under interventions—much like a causal model.

The problem of evaluating policies under distributional shifts has been studied extensively in causal transportability (CT) theory (Bareinboim & Pearl, 2016; Bellot & Bareinboim, 2022). CT aims to provide necessary and sufficient conditions for policy evaluation under known domain shifts when all assumptions on the data generating process (i.e. inductive biases) can be expressed as constraints on causal structure (Bareinboim & Pearl, 2016). However, deep learning algorithms can exploit a much larger set of inductive biases (Neyshabur et al., 2014; Battaglia et al., 2018; Rahaman et al., 2019; Goyal & Bengio, 2022) which in many real-world tasks may be sufficient to identify low regret policies without requiring causal knowledge. Thus, CT does not imply that agents must learn causal models in order to generalise unless we assume agents only use causal assumptions to begin with, which would be proof by assumption. See Appendix G for further discussion.

A similar result to Theorems 1 and 2 is the causal hierarchy theorem (CHT) (Bareinboim et al., 2022; Ibeling & Icard, 2021), which shows that observational data is almost always *insufficient* for identifying all causal relations between environment variables, whereas our results state that the set of optimal policies is almost always sufficient to identify all causal relations. In Section 3.2 we discuss the similarities between these theorems, and in Appendix G we discuss their differences.

## 6 CONCLUSION

Causal reasoning is foundational to human intelligence, and has been conjectured to be necessary for achieving human level AI (Pearl, 2019). In recent years, this conjecture has been challenged by the development of artificial agents capable of generalising to new tasks and domains without explicitly learning or reasoning on causal models. And while the necessity of causal models for solving causal inference tasks has been established (Bareinboim et al., 2022), their role in decision tasks such as classification and reinforcement learning is less clear.

We have resolved this conjecture in a model-independent way, showing that any agent capable of robustly solving a decision task must have learned a causal model of the data generating process, regardless of how the agent is trained or the details of its architecture. This hints at an even deeper connection between causality and general intelligence, as this causal model can be used to find policies that optimise any given objective function over the environment variables. By establishing a formal connection between causality and generalisation, our results show that causal world models are a necessary ingredient for robust and general AI.

**Acknowledgements.** We would like to thank Alexis Bellot, Damiano Fornasiere, Pietro Greiner, James Fox, Matt MacDermott, David Reber, David Watson and Philip Bachman for their helpful discussions and comments on the manuscript.

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

# A PRELIMINARIES

## A.1 SETUP AND ASSUMPTIONS

The environment is described by a set of random variables $\boldsymbol{C} = \{C_1, C_2, \ldots, C_N\}$, which in combination with the decision $D$ and utility nodes $U$ define the state space for the CID $\boldsymbol{V} = \boldsymbol{C} \cup \{D, U\}$. In out notation individual variables $C_i \in \boldsymbol{C}$ are given indexes, whereas set of variables are indexless and bold, and we use $\boldsymbol{V} = \boldsymbol{v}$ as short hand for the joint state of the variables in a set $\boldsymbol{C}$. The joint probability distribution $P(\boldsymbol{C} = \boldsymbol{c}, D = d, U = u)$ describes the statistical relations between environment variables. Bayesian networks factorise joint probability distributions according to a graph $G$ (Pearl, 2009).

**Definition 1** (Bayesian networks). *A Bayesian network $M = (G, P)$ over a set of variables $\boldsymbol{V} = \{V_1, \ldots, V_n\}$ is a joint probability distribution $P(\boldsymbol{V})$ that factors according to a directed acyclic graph (DAG) $G$, i.e. $P(V_1, \ldots, V_n) = \prod_{i=1}^{n} P(V_i \mid \boldsymbol{Pa}_{V_i})$, where $\boldsymbol{Pa}_{V_i}$ are the parents of $V_i$ in $G$.*

The distributions and statistical relationships between variables may change as a result of external *interventions* applied to a system. *Hard* interventions set a subset $\boldsymbol{C}' \subseteq \boldsymbol{C}$ of the variables to particular values $\boldsymbol{c}'$, denoted $\mathrm{do}(\boldsymbol{C}' = \boldsymbol{c}')$ or $\mathrm{do}(\boldsymbol{c}')$. Naively, one joint probability distribution $P_{\mathrm{do}(\boldsymbol{c}')}$ would be needed to describe the updated relationship under each possible intervention $\mathrm{do}(\boldsymbol{c}')$. Fortunately, all interventional distributions can be derived from a single Bayesian network, if $G$ matches the causal structure of the environment (i.e. has an edge $V_i \to V_j$ whenever an intervention on $V_i$ directly influences the value of another variable $Y$, and lacks unmodeled confounders; Spirtes et al., 2000; Pearl, 2009). When this holds, we call the Bayesian network *causal* and $G$ a *causal graph*. With respect to the causal graph $G$ we denote the direct causes (parents) of $V_i$ as $\mathbf{Pa}_i$, the set of all causes (ancestors) $\mathbf{Anc}_i$, and the variables that $V_i$ directly causes (children) $\mathbf{Ch}_i$ and descendants $\mathbf{Desc}_i$ as the set of all downstream variables. Note in particular that $\mathbf{Anc}_i$ and $\mathbf{Desc}_i$ refer to *proper* ancestors and descendants, i.e. $V_i \notin \mathbf{Anc}_i$ and $V_i \notin \mathbf{Desc}_i$. We denote a causal Bayesian network (CBN) as $M = (P, G)$ where $P$ is the joint and $G$ is the directed acyclic graph (DAG) describing the causal structure of the environment. Further, the interventional distribution $P_{\mathrm{do}(v')}$ is given by the truncated factorisation

$$P_{\mathrm{do}(v')}(\boldsymbol{v}) = \begin{cases} \prod_{i:v_i \notin \boldsymbol{v}'} P(v_i \mid \mathbf{pa}_{v_i}) & \text{if } \boldsymbol{v} \text{ consistent with } \boldsymbol{v}' \\ 0 & \text{otherwise.} \end{cases}$$

Equivalently, the effect of interventions can be computed by adding an extra node $\hat{X}$ and edge $\hat{V}_i \to V_i$ for each node $V_i \in V$ (Correa & Bareinboim, 2020; Dawid, 2002). Intervening on $V_i$ then corresponds to conditioning on $\hat{V}_i$ in the extended graph. More general, soft interventions $\sigma = P'(V_i \mid \mathbf{Pa}_i^*)$ replace the conditional probability distribution for $V_i$ with a new one, possibly using a new parent set $\mathbf{Pa}_o^*$ as long as no cycles are introduced in the graph (Correa & Bareinboim, 2020). The modified environment is denoted $M(\sigma)$.

General soft interventions cannot be defined without prior knowledge of the causal graph $G$. For example, the soft intervention $\sigma_Y = P'(y \mid x)$ is incompatible with the causal structure $Y \to X$ as it would introduce a causal cycle, and so an agent's policy may not be well defined with respect to this intervention. We therefore focus our theoretical analysis on a subset of the soft interventions, *local interventions*, that can be implemented without assuming knowledge of $G$.

**Definition 2** (Local interventions). *Local intervention $\sigma$ on $V_i \in \boldsymbol{V}$ involves applying a map to the states of $V_i$ that is not conditional on any other endogenous variables, $v_i \mapsto f(v_i)$. We use the notation $\sigma = do(V_i = f(v_i))$ (variable $V_i$ is assigned the state $f(v_i)$). Formally, this is a soft intervention on $V_i$ that transforms the conditional probability distribution as,*

$$P(v_i \mid \boldsymbol{pa}_i; \sigma) = \sum_{v_i' : f(v_i') = v_i} P(v_i' \mid \boldsymbol{pa}_i) \tag{1}$$

**Example:** Fixing the value of a variable (hard intervention) is a local intervention as $\mathrm{do}(V_i = v_i') = \mathrm{do}(V_i = f(v_i))$ where $f(v_i) = v_i'$.

**Example:** Translations are local interventions as $\mathrm{do}(V_i = v_i + k) = \mathrm{do}(V_i = f(v_i))$ where $f(v_i) = v_i + k$. This includes changing the position of objects in RL environments (Shah et al., 2022).

**Example:** Logical NOT operation $X \to \neg X$ for Boolean $X$

We also consider mixtures of interventions, which can also be described without knowledge of $G$.

**Definition 3** (Mixtures of interventions). *A mixed intervention $\sigma^* = \sum_i p_i \sigma_i$ for $\sum p_i = 1$ performs intervention $\sigma_i$ with probability $p_i$. Formally, $P(\boldsymbol{v} \mid \sigma^*) = \sum_i p_i P(\boldsymbol{v} \mid \sigma_i)$.*

**Example:** Adding Gaussian noise is a mixture over local operations (translations) $\sigma_\epsilon = \mathrm{do}(X = X + \epsilon)$ where $\epsilon \sim \mathcal{N}(0,1)$.

In common to most decision making tasks such as prediction, classification, and reinforcement learning, is that a decision should be outputted based on some information to optimise some objective. The exact terms vary: decisions are sometimes called outputs, actions, predictions, or classifications; information is sometimes called features, context, or state; and objectives are sometimes called utility functions or loss functions. However, all of these setups can be described within the causal influence diagram (CID) framework (Howard & Matheson, 2005; Everitt et al., 2021). CIDs are causal Bayesian networks where the variables are divided into decision $D$, utility $U$, and chance variables $V$, and no conditional probability distribution is specified for the decision variables. The task of the agent is to select the distribution $\pi = P(D = d \mid \mathbf{Pa}_D = \mathbf{pa}_D)$, also known as the policy or decision rule. An optimal policy $\pi^*$ is defined as a policy $\pi^*$ that maximizes the expected value of the utility $\mathbb{E}_{\pi^*}[U]$.

**Definition 4** (Causal influence diagram). *A (single-decision, single-utility) causal influence diagram (CID) is a CBN $M = (G, P)$ where the variables $\boldsymbol{V}$ are partitioned into decision, utility, and chance variables, $\boldsymbol{V} = (\{D\}, \{U\}, \boldsymbol{C})$. The utility variable is a real-valued function of its parents, $U(\boldsymbol{pa}_U)$.*

By convention, decision nodes are drawn square, utility nodes diamond, and chance nodes round. The parents of $D$, $\mathbf{Pa}_D$, can be interpreted as the information the decision is allowed to depend on, and are depicted as dashed lines. See Figure 5 for an example. In the following we restrict our attention to a class of CIDs we refer to as 'unmediated decision tasks', where where the agent's decision does not causally influence any chance variables that go on to influence the utility. This simplifies our theoretical analysis, although it is likely that our results extend to the general case.

**Assumption 1** (Unmediated decision task). *$\boldsymbol{Desc}_D \cap \boldsymbol{Anc}_U = \emptyset$.*

Examples of unmediated decision tasks include all standard classification and regression tasks, and generative AI tasks where the output is not included in the training set. For example, in classification typically the choice of label does not influence the data generating process. Problems that are mediated rather than unmediated decision tasks includes most control and reinforcement learning tasks, where the agent's decision is an action that influences the state of the environment. Furthermore, we will focus on non-trivial unmediated decision tasks i.e. where $U \in \mathbf{Ch}_D$, as the case $\mathbf{Ch}_D = \emptyset$ describes trivial decision tasks (the agent's action does not influence the utility). Figure 5 is an example of a non-trivial unmediated decision task.

In transfer learning we are typically interested in problems where generalising from the source to target domain(s) is non-trivial, and in the trivial case we cannot expect agents to have to learn anything about their environments in order to generalise. If this is not the case, then generalising under distributional shifts is trivial. Therefore we restrict our attention to decision tasks where the distribution of the environment is relevant to the agent when determining its policy. Specifically, we say that a decision task is *domain independent* if there exists a single policy that is optimal for all choices of environment distribution $P(\boldsymbol{C} = \boldsymbol{c})$.

**Assumption 2** (Domain dependence). *There exists $P(\boldsymbol{C} = \boldsymbol{c})$ and $P'(\boldsymbol{C} = \boldsymbol{c})$ compatible with $M$ such that $\pi^* = \arg\max_\pi \mathbb{E}_P^\pi[U]$ implies $\pi^* \neq \arg\max_\pi \mathbb{E}_{P'}^\pi[U]$.*

**Lemma 1.** *Domain dependence implies that;*

*i) There exists no $d \in dom(D)$ such that $d \in \arg\max_d U(d, c)\ \forall \boldsymbol{c} \in dom(C)$.*

*ii) $\boldsymbol{Pa}_D \subsetneq \boldsymbol{Anc}_U$*

*iii) $D \in \boldsymbol{Pa}_U$*

*Proof.* i) For any $P'$ we have $\mathbb{E}_{P'}[U \mid \mathrm{do}(D = d), \mathbf{pa}_D] = \sum_{\boldsymbol{c}} P'(\boldsymbol{C}_d = \boldsymbol{c} \mid \mathbf{pa}_D)U(d, c)$ $= \sum_{\boldsymbol{c}} P'(\boldsymbol{C} = \boldsymbol{c} \mid \mathbf{pa}_D)U(d, c)$ where we have used $\mathbf{Desc}_D \cap \mathbf{Anc}_U = \emptyset$. Therefore if $\exists\ d^*$

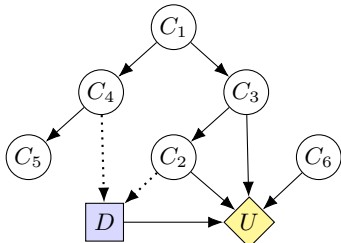

Figure 5: The CID for an unmediated decision task, where $D$ has no causal influence on the environment state $C$. Our main theorem implies that an agent that is robust to distributional shifts on $C$ must learn the CBN over $\mathbf{Anc}_U = \{C_1, C_2, C_3, C_4, C_6\}$, noting that $C_5 \notin \mathbf{Anc}_U$. $C_4$ is an example of a variable that is only an ancestor of $U$ via $D$ and so has no direct causal effect on the utility, but is still relevant to the decision task as it is a proxy for $C_1$ which is a cause of $U$. $C_6$ is a cause of $U$ but not of $D$, and naively one might assume that distributional shifts on $C_6$ cannot influence the agent's decision. However, the optimal policy can change under distributional shifts on $C_6$ as these effect the utility, and hence the agent will have to learn a CBN including $C_6$ if it is to be robust to shifts on $C_6$.

s.t. $d^* = \arg\max_d U(d, \boldsymbol{c}) \ \forall \ \boldsymbol{C} = \boldsymbol{c}$ then $\mathbb{E}_{P'}[U \mid \mathrm{do}(D = d^*), \mathbf{pa}_D] \geq \sum_{\boldsymbol{c}} P'(\boldsymbol{C} = \boldsymbol{c} \mid \mathbf{pa}_D) U(d', \boldsymbol{c}) \mathbb{E}_{P'}[U \mid \mathrm{do}(D = d), \mathbf{pa}_D] \ \forall \ d \neq d^*$, and so $D = d^*$ is optimal for all $P'(\boldsymbol{C} = \boldsymbol{c})$ and we violate domain dependence.

ii) As $D \in \mathbf{Pa}_U$ (iii), then $\mathbf{Pa}_U \subseteq \mathbf{Anc}_U$. If $\mathbf{Anc}_U = \mathbf{Pa}_D$ then $\mathbb{E}_P[u \mid d, \mathbf{pa}_D] = U(d, \mathbf{pa}_D)$ which is independent of $P(\boldsymbol{C} = \boldsymbol{c})$, and hence there is a single optimal policy for all $P$ and we violate domain dependence.

iii) If $D \notin \mathbf{Anc}_U$ then the CID is trivial, in the sense that $\mathbb{E}[U \mid \mathrm{do}(D = d)] = \mathbb{E}[U]$, and hence all decisions are optimal for all distributions $P(\boldsymbol{C})$, which violates domain dependence (Assumption 2). Therefore $D \in \mathbf{Anc}_U$ which with $\mathbf{Desc}_D \cap \mathbf{Anc}_U = \emptyset$ implies $D \in \mathbf{Pa}_U$. $\qquad\square$

## A.2 PARAMETERISATION OF CIDS

The joint distribution $P$ is defined for all environment variables $\boldsymbol{C}$, and the CID is defined by the parameters for $P(\boldsymbol{C})$ and $U(\mathbf{Pa}_U)$. We restrict our attention to $\boldsymbol{C}$ that are categorical, and without loss of generality we label states $c_i = 0, 1, \ldots, \dim_i - 1$ where $\dim_i$ is the dimension of variable $C_i$. Firstly, the joint $P(\boldsymbol{C})$ is parameterised by the conditional probability distributions (CPDs) in the Markov factorization with respect to $G$, $\theta_P = \{P(c_i \mid \mathbf{pa}_i) \ \forall \ c_i \in \{0, \ldots, \dim_i - 2\}, \mathbf{pa}_i \in \mathbf{Pa}_i, C_i \in \boldsymbol{C}$. Note that the CPDs $p(C_i = \dim_i - 1 \mid \mathbf{pa}_i)$ are not included in $\theta_P$ as they are fully constrained by normalization $P(C_i = \dim_i - 1 \mid \mathbf{pa}_i) = 1 - \sum_{j=0}^{\dim_i - 1} P(c_i \mid \mathbf{pa}_i)$. Secondly, the utility function is simply parameterised by its value given the state of its parents $\theta_U = \{U(\mathbf{pa}_U) \ \forall \ \mathbf{Pa}_U = \mathbf{pa}_U\}$. For simplicity we work with the normalized utility function,

$$U(\mathbf{pa}_U) \rightarrow \frac{U(\mathbf{pa}_U) - \min_{\mathbf{pa}'_U} U(\mathbf{Pa}_U = \mathbf{pa}'_U)}{\max_{\mathbf{pa}'_U} U(\mathbf{Pa}_U = \mathbf{pa}'_U) - \min_{\mathbf{pa}'_U} U(\mathbf{Pa}_U = \mathbf{pa}'_U)} \tag{2}$$

with values between 0 and 1. Noting that as this is a positive affine transformation of the utility function the set of optimal policies invariant, and we can re-scale regret bounds accordingly. Let $\theta_M$ denote the set of all parameters for the CID, $\theta_M = \theta_P \cup \theta_U$, and note that the elements of $\theta_M$ in the $[0, 1]$ interval and are logically independent, i.e. we can independently choose any $[0, 1]$ value for each parameter and this defines a valid parameterization of the CID for the baseline environment. In the following when we refer to 'the parameters $P, U$' we are referring to $\theta_M$.

We follow the method outlined in (Meek, 2013) to prove that certain constraints on $P, U$ hold 'for almost all $P, U$' and hence for almost all decision tasks. This involves converting a given constraint into polynomial equations over $\theta_M$ and applying the following Lemma,

**Lemma 2** (Okamoto, 1973). *The solutions to a (nontrivial) polynomial are Lebesgue measure zero over the space of the parameters of the polynomial.*

A polynomial in $n$ variables is non-trivial (not an identity) if not all instantiations of the $n$ variables are solutions of the polynomial. For example, the equation $\text{poly}(\theta_M) = 0$ is trivial if and only if all coefficients of the polynomial expression $\text{poly}(\theta_M)$ are zero. Therefore, any constraint on $P, U$ that can be converted into a polynomial equation over $\theta_M$ must either hold for all $\theta_M$ or for a Lebesgue measure zero subset of instantiations of $\theta_M$.

Operationally, this means that if we have any smooth distribution over the parameter space (for example, describing the distribution of environments we expect to encounter), the probability of drawing an environment from this distribution for which the condition does not hold is 0.

### A.3 DISTRIBUTIONAL SHIFTS & POLICY ORACLES

In the derivation of our results we restrict out attention to distributional shifts that can be modelled as (soft) interventions on the data generating process. We note that by Reichenbach's principle (Reichenbach, 1956), which states that all statistical associations are due to underlying causal structures, we can assume the existence of a causal data generating process that can be described in terms of a CBN $M = (P, G)$. Therefore there is a causal factorization of the joint $P(\boldsymbol{C} = \boldsymbol{c}) = \prod_i P(c_i \mid \mathbf{Pa}_i)$. By allowing for mixtures of interventions, we can reach any distribution over $\boldsymbol{C}$, which can be seen trivially by noting that we can perform a soft intervention to achieve any deterministic distribution $P(\boldsymbol{C} = \boldsymbol{c}) = \delta(\boldsymbol{C} = \boldsymbol{c}')$, and then take a mixture over these deterministic distributions to achieve an arbitrary distribution over $\boldsymbol{C}$. The set of distributions that cannot be generated by interventions include those that change the set of variables $\boldsymbol{V}$ including the decision and utility variables, and introducing selection biases (which are causally represented with the introduction of additional nodes that are conditioned on Bareinboim & Pearl, 2012a). For further discussions on the relation between distributional shifts and interventions see Schölkopf et al. (2021); Meinshausen (2018).

In the following proofs we use *policy oracles* to formalise knowledge of regret-bounded behaviour under distributional shifts.

**Definition 5** (Policy oracle). *A policy oracle for a set of interventions $\Sigma$ is a map $\Pi_\Sigma^\delta : \sigma \mapsto \pi_\sigma(d \mid \boldsymbol{pa}_D) \ \forall \ \sigma \in \Sigma$ where $\Sigma$ is a set of domains. It is $\delta$-optimal if $\pi_\sigma(d \mid \boldsymbol{pa}_D)$ achieves an expected utility $\mathbb{E}^{\pi_\sigma}[U] \geq \mathbb{E}^{\pi^*}[U] - \delta$ in the CID $M(\sigma)$ where $\delta \geq 0$.*

Here $\delta$ is the regret upper bound, which is satisfied under all distributional shifts $\sigma \in \Sigma$. We refer to $\delta$-optimal policy oracles for $\delta = 0$ as optimal policy oracles. For the proof of our main result we restrict our attention to policy oracles with $\Sigma$ that includes mixtures over all local interventions (def. 2).

Note that the policy oracle specifies only what policy the agent returns in a distributionally shifted environment $M(\sigma)$. It does not specify how this policy is generated, which will depend on the specific setup. For example, in domain generalisation that agent typically receives no additional data from the target domains, and is expected to produce a policy (decision boundary) that achieves a low regret across all target domains. On the other hand in domain adaptation and few shot learning, the agent is provided with some new data from each target domain with which to adjust its policy. As we hope to accommodate all of these perspectives we specify only the agent's policy, not the data used to generate it. This is discussed further in Section 3.2.

**What distributional shifts do we consider?** In our proofs, we assume the agent is robust to any domain shifts that can be described as a mixture of local interventions on the environment variables $\boldsymbol{C}$. We do not consider interventions that change the utility $U$ or the agent's decision $D$, though we do include dropping inputs to the policy (masking) $\mathbf{Pa}_D \to \mathbf{Pa}'_D \subseteq \mathbf{Pa}_D$ as local interventions.

## B APPENDIX: SIMPLIFIED PROOF

In this section we outline the proof of Theorem 1 for a simple binary decision task with binary latent variables. As mentioned in Section 4, the method used to identify the CBN in Theorem 1 can be viewed as an algorithm for learning the CBN over latent variables by observing the policy of a regret-bounded agent under various distributional shifts. To demonstrate this, in Appendix F we use an implementation of the algorithm on randomly generated CIDs, showing empirically that we can

learn the underlying CBN in this way, and explore how the agent's regret bound affects the accuracy of the learned CBN.

Consider the CID in Figure 6, describing a binary decision task $D \in \{0, 1\}$ with two binary latent variables $X, Y \in \mathbf{Pa}_U$.

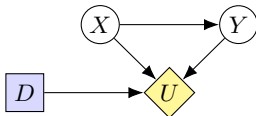

Figure 6: Example CID describing a context-free mutli-armed bandit with binary latent variables $X, Y$.

Consider an agent that selects a policy $\pi_D$ such that it maximises the expected utility. That is, the CID describes a context-free bandit problem, where $X, Y$ are latent variables that influence the arm values $\mathbb{E}[u \mid d] = \sum_{x,y} P(x, y)U(x, y, d)$.

Our aim is to learn this CID given only knowledge of the agent's policy under distributional shifts, and knowledge that it satisfies a regret bound. We assume knowledge of i) the set of chance variables $\mathbf{C} = \{X, Y\}$, ii) the utility function $U(d, x, y)$, and iii) the policy $\pi_D(\sigma)$ under distributional shifts $\sigma$ (other variables $(U, X, Y)$ are unobserved). To learn the CID the aim is therefore to learn the parameters of the joint distribution over latents $P(x, y)$ and the unknown causal structure. As we know the utility function we know $D, X, Y \in \mathbf{Pa}_U$, and by assuming the CID is unmediated (Assumption 1) we know $X, Y \notin \mathbf{Desc}_D$. Likewise the decision task is context free hence $D \notin \mathbf{Desc}_X \cup \mathbf{Desc}_Y$. Hence the only unknown causal structure is the DAG over the latent variables $\mathbf{C} = \{X, Y\}$.

The expected utility difference between $D = 0$ and $D = 1$ following a hard intervention on $X$ is given by

$$\mathbb{E}[u \mid D = 0; \text{do}(X = 0)] - \mathbb{E}[u \mid D = 1; \text{do}(X = 0)] = \sum_y P(Y_{X=0} = y)[U(0, 0, y) - U(1, 0, y)]$$

(3)

$$= P(Y_{X=0} = 0)[U(0, 0, Y = 0) - U(1, 0, 0)] + (1 - P(Y_{X=0} = 0))[U(0, 0, 1) - U(1, 0, 1)]$$

(4)

As we know $U(d, x, y)$ we can therefore identify $P(Y_{X=0} = 0)$ if we can identify this expected utility difference. We do this using the agent's policy under distributional shifts, and in this simple case we can restrict our attention to hard interventions. Following the steps outlined in Lemma 4, domain dependence insures that we can identify a hard intervention $\sigma_2 = \text{do}(X = x', Y = y')$ that results in a different optimal policy to the optimal policy under $\sigma_1 = \text{do}(X = 0)$. For a mixture of these two interventions $\sigma_3 = q\sigma_1 + (1 - q)\sigma_2$ the expected utility is $\mathbb{E}[u \mid d, \sigma_3] = q\mathbb{E}[u \mid d, \sigma_1] + (1 - q)\mathbb{E}[u \mid d, \sigma_2]$. This is a linear function with respect to $q$, and for $q = 1$ the optimal decision ($d_1$) is different than for $q = 0$ ($d_2 \neq d_1$). Therefore, there is a single indifference point $q_{\text{crit}}$ for which both decisions are optimal. It is simple to show that this indifference point is given by,

$$q_{\text{crit}} = \left(1 - \frac{\mathbb{E}[u \mid D = d_1; \text{do}(X = 0)] - \mathbb{E}[u \mid D = d_2; \text{do}(X = 0)]}{U(d_1, x', y') - U(d_2, x', y')}\right)^{-1}$$

(5)

$D = d_1$ is optimal for $q \leq q_{\text{crit}}$ and $D = d_2$ is optimal for $q \geq q_{\text{crit}}$. We can estimate $q_{\text{crit}}$ by randomly sampling values of $q$ uniformly over $[0, 1]$ and observing the optimal decision under the resulting mixed intervention (Algorithm 1). That is, $q_{\text{crit}}$ is the probability that $D = d_1$ is returned by the policy oracle for a randomly sampled $q$. In this way we learn $q_{\text{crit}}$ and as we know $U(d, x, y)$ we can identify the expected utility difference under $\text{do}(X = 0)$ in the numerator of Equation (5) and so identify $P(Y_{X=0} = 0)$.

Similarly we identify $P(Y_{X=1} = 0), P(X_{Y=0} = 0)$ and $P(X_{Y=1} = 0)$, which encode both the causal relation between $X$ and $Y$ (e.g. there is a directed path from $X$ to $Y$ if and only if $P(Y_{X=0}) \neq P(Y_{X=1})$ for almost all CBNs), and determine the parameters of the CBN as $P(C_i = c_i \mid \text{do}(\mathbf{C} \setminus C_i)) = P(C_i = c_i \mid \mathbf{Pa}_i = \mathbf{pa}_i)$.

## C  PROOF OF THEOREM 1

In this appendix we prove Theorem 1. For an informal overview of the proof see Appendix B.

First, we show that for a given distributional shift $\sigma$, for almost all $P, U$ there is a single optimal decision. While this is not necessary for our proof, it simplifies our analysis. And as our main theorem holds for almost all $P, U$, we can include any finite number of independent conditions that hold for almost all $P, U$ without strengthening this condition, as the union of Lebesgue measure zero sets is Lebesgue measure zero.

**Lemma 3.** *For any given local intervention $\sigma$ there is a single deterministic optimal policy for almost all $P, U$.*

*Proof.* Following intervention $\sigma$ two decisions $d, d'$ are simultaneously optimal in context $\mathbf{pa}_D$ if,

$$\mathbb{E}[u \mid \mathbf{pa}_D, \mathrm{do}(D = d); \sigma] = \mathbb{E}[u \mid \mathbf{pa}_D, \mathrm{do}(D = d'); \sigma] \tag{6}$$

Let $\boldsymbol{Z} = [\mathbf{Anc}_U \setminus \mathbf{Pa}_D]$ and $\boldsymbol{X} = \mathbf{Pa}_U \setminus \{D\}$. Noting that

$$\mathbb{E}[u \mid \mathbf{pa}_D, \mathrm{do}(D = d); \sigma] = \sum_{\boldsymbol{z}} U(d, \boldsymbol{x}) P(\boldsymbol{z}, \mathbf{pa}_D \mid \mathrm{do}(D = d); \sigma) / P(\mathbf{pa}_D \mid \mathrm{do}(D = d); \sigma) \tag{7}$$

and that $P(\mathbf{pa}_D \mid \mathrm{do}(D = d); \sigma) = P(\mathbf{pa}_D; \sigma)$ and $P(\boldsymbol{z}, \mathbf{pa}_D \mid \mathrm{do}(D = d); \sigma) = P(\boldsymbol{z}, \mathbf{pa}_D; \sigma)$ which follows from $\mathbf{Desc}_D \cap \mathbf{Anc}_U = \emptyset$, we can multiple both sides of equation 6 with $P(\mathbf{pa}_D; \sigma)$ giving,

$$\sum_{\boldsymbol{z}} U(d, \boldsymbol{x}) P(\boldsymbol{z}, \mathbf{pa}_D; \sigma) = \sum_{\boldsymbol{z}} U(d', \boldsymbol{x}) P(\boldsymbol{z}, \mathbf{pa}_D; \sigma) \tag{8}$$

and

$$\sum_{\boldsymbol{z}} [U(d, \boldsymbol{x}) - U(d', \boldsymbol{x})] P(\boldsymbol{z}, \mathbf{pa}_D; \sigma) = 0 \tag{9}$$

Let $\sigma = \mathrm{do}(v_1 = f_1(v_1), \ldots, v_N = f_N(v_N))$. The joint $P(\boldsymbol{z}, \mathbf{pa}_D; \sigma) = \prod_i P(c_i \mid \mathbf{pa}_i; \sigma)$ is polynomial, and the local interventions $P(c_i \mid \mathbf{pa}_i; \sigma) = \sum_{c_i' : f_i(c_i') = c_i} P(c_i' \mid \mathbf{pa}_i)$ keep it polynomial. Therefore equation 9 is a polynomial equation over the model parameters, and is certain to be non-trivial as $d \neq d'$. Therefore by Lemma 2 for almost all $P, U$ equation 9 is not satisfied, and as there are a finite number of decisions this implies that for almost all $P, U$ there is a single optimal decision for a given $\sigma, \mathbf{pa}_D$ and hence a single optimal policy. $\qquad\square$

Next, we detail how a policy oracle can be used to identify a specific causal query in the shifted environment $M(\sigma)$, that we will later use to identify the model parameters.

**Lemma 4.** *Using an optimal policy oracle $\Pi_\Sigma^*$ where $\Sigma$ includes all mixtures of local interventions on $\boldsymbol{C}$ including masking inputs $\boldsymbol{Pa}_D' \subseteq \boldsymbol{Pa}_D$, then for any given $\boldsymbol{Pa}_D' = \boldsymbol{pa}_D'$ such that $\boldsymbol{Pa}_D' \cap \boldsymbol{Pa}_U = \emptyset$ we can identify $\sum_z P(\boldsymbol{C} = \boldsymbol{c}; \sigma)[U(d, \boldsymbol{c}) - U(d', \boldsymbol{c})]$, for $d$ and $d'$ where $d \neq d'$ and $\boldsymbol{Z} = \boldsymbol{C} \setminus \boldsymbol{Pa}_D'$.*

*Proof.* By Lemma 3 for almost all $P, U$ there is a single optimal decision following the shift $\sigma$. Let $d_1 = \arg\max_d \mathbb{E}[u \mid \mathrm{do}(D = d), \mathbf{pa}_D'; \sigma]$ where $d_1 = \pi^*(\sigma)$. We can identify $d_1$ by querying the policy oracle with $\sigma$.

Consider a hard intervention on all $C_i \in \boldsymbol{C}$, $\sigma' := \mathrm{do}(c_1', c_2', \ldots, c_N')$ where for all $C_i \in \mathbf{Pa}_D'$ we set $C_i = c_i$ to be the same state as in observation $\mathbf{Pa}_D' = \mathbf{pa}_D'$. The expected utility under this intervention is $\mathbb{E}[u \mid \mathrm{do}(D = d), \mathbf{pa}_D'; \sigma'] = U(d, \boldsymbol{x}')$ where $\boldsymbol{X} = \mathbf{Pa}_U \setminus \{D\}$ (and we have that $D \in \mathbf{Pa}_U$ from Lemma 1 iii)).

Next we show that there is a choice of hard intervention $\sigma'$ such that the policy oracle must return different optimal decisions in the context $\mathbf{Pa}_D' = \mathbf{pa}_D'$ for $\sigma'$ and $\sigma$. As $\mathbf{Pa}_D' \cap \mathbf{Pa}_U = \emptyset$ then we are free to choose any $X = x'$ and the resulting $\sigma'$ will be compatible with the evidence $\mathbf{Pa}_D' = \mathbf{pa}_D'$. Note that by Lemma 1 i) $\exists \, \boldsymbol{X} = \boldsymbol{x}'$ s.t. $d_1 \neq \arg\max_d U(d, x')$, else $D = d_1$ is optimal for all $\boldsymbol{X} = \boldsymbol{x}$ which violates domain dependence. We can determine this $\boldsymbol{X} = \boldsymbol{x}'$ given the utility function and $d_1$. Let $d_2 = \arg\max_d U(d, \boldsymbol{x}')$ and $\sigma' = \mathrm{do}(c_1', c_2', \ldots, c_N')$ be the hard intervention for which $\boldsymbol{X} = \boldsymbol{x}'$ and $\mathbf{Pa}_D' = \mathbf{pa}_D'$.

Consider the joint distribution over $C$ under the mixed local intervention $\tilde{\sigma}(q) = q\sigma + (1-q)\sigma'$,

$$P(\boldsymbol{C} = \boldsymbol{c} \mid \mathrm{do}(D = d); \tilde{\sigma}(q)) = P(\boldsymbol{C} = \boldsymbol{c}; \tilde{\sigma}(q)) \tag{10}$$

$$= qP(\boldsymbol{C} = \boldsymbol{c}; \sigma) + (1-q)P(\boldsymbol{C} = \boldsymbol{c}; \sigma') \tag{11}$$

where in the first line we have used $\mathbf{Ch}_D = \{U\}$ to drop the intervention. Note that $\boldsymbol{Z} = \boldsymbol{C} \backslash \mathbf{Pa}_D \neq \emptyset$ by Lemma 1 i). The expected utility is given by,

$$\mathbb{E}[u \mid \mathbf{pa}_D, \mathrm{do}(D = d); \tilde{\sigma}(q)] = \sum_{\boldsymbol{z}} P(\boldsymbol{Z} = \boldsymbol{z} \mid \mathbf{pa}_D, \mathrm{do}(D = d); \tilde{\sigma}(q)) U(d, \boldsymbol{x}) \tag{12}$$

$$= \sum_{\boldsymbol{z}} \frac{P(\boldsymbol{C} = \boldsymbol{c} \mid \mathrm{do}(D = d); \tilde{\sigma}(q))}{P(\mathbf{pa}_D \mid \mathrm{do}(D = d); \tilde{\sigma}(q))} U(d, \boldsymbol{x}) \tag{13}$$

$$= \frac{1}{P(\mathbf{pa}_D; \tilde{\sigma}(q))} \sum_{\boldsymbol{z}} P(\boldsymbol{C} = \boldsymbol{c}; \tilde{\sigma}(q)) U(d, \boldsymbol{x}) \tag{14}$$

$$= \frac{1}{P(\mathbf{pa}_D; \tilde{\sigma}(q))} \sum_{\boldsymbol{z}} qP(\boldsymbol{C} = \boldsymbol{c}; \sigma) U(d, \boldsymbol{x}) + (1-q)P(\boldsymbol{C} = \boldsymbol{c}; \sigma') U(d, \boldsymbol{x}') \tag{15}$$

Note that for $q = 1$ the optimal decision is $d_1$ and for $q = 0$ the optimal decision returned by the policy oracle belongs to the set $\{d \text{ s.t. } d = \arg\max_d U(d, \boldsymbol{x}')\}$ which does not contain $d_1$. Furthermore, the argmax of equation 15 with respect to $d$ is a piecewise linear function with domain $q \in [0, 1]$. Therefore there must be some $q = q_{\mathrm{crit}}$ that is the smallest value of $q$ such that for $q < q_{\mathrm{crit}}$ the policy oracle returns an optimal decision in the set $\{d \text{ s.t. } d = \arg\max_d U(d, \boldsymbol{x}')\}$ and for $q \geq q_{\mathrm{crit}}$ the optimal decision is not in this set. The value of $q_{\mathrm{crit}}$ is given by $\mathbb{E}[u \mid \mathbf{pa}_D, \mathrm{do}(D = d); \tilde{\sigma}(q_{\mathrm{crit}})] = 0$, which by equation 15 is,

$$q_{\mathrm{crit}} \sum_{\boldsymbol{z}} P(\boldsymbol{C} = \boldsymbol{c}; \sigma)[U(d_2, \boldsymbol{x}) - U(d_3, \boldsymbol{x})] + (1 - q_{\mathrm{crit}})[U(d_2, \boldsymbol{x}') - U(d_3, \boldsymbol{x}')] = 0 \tag{16}$$

where $d_2 \in \{d \text{ s.t. } d = \arg\max_d U(d, \boldsymbol{x}')\}$ and $d_3 \notin \{d \text{ s.t. } d = \arg\max_d U(d, \boldsymbol{x}')\}$. This yields the following expression for $q_{\mathrm{crit}}$,

$$q_{\mathrm{crit}} = \left(1 - \frac{\sum_{\boldsymbol{z}} P(\boldsymbol{C} = \boldsymbol{c}; \sigma)[U(d_2, \boldsymbol{x}) - U(d_3, \boldsymbol{x})]}{U(d_2, \boldsymbol{x}') - U(d_3, \boldsymbol{x}')}\right)^{-1} \tag{17}$$

where we have used $\sum_{\boldsymbol{z}} P(\boldsymbol{C} = \boldsymbol{c}; \sigma')[U(d_2, \boldsymbol{c}) - U(d_3, \boldsymbol{c})] = U(d_2, \boldsymbol{x}') - U(d_3, \boldsymbol{x}')$. We can determine $\sum_{\boldsymbol{z}} P(\boldsymbol{C} = \boldsymbol{c}; \sigma)[U(d_2, \boldsymbol{x}) - U(d_3, \boldsymbol{x})]$ given $q_{\mathrm{crit}}$ and the utility function $U(d, \boldsymbol{x})$.

Finally, we describe a Algorithm 1 (below) that uses a policy oracle for the Monte Carlo estimation of $q_{\mathrm{crit}}$, which can be used to determine $\sum_{\boldsymbol{z}} P(\boldsymbol{C} = \boldsymbol{c}; \sigma)[U(d_2, \boldsymbol{c}) - U(d_3, \boldsymbol{c})])$ in the asymptotic limit $N \to \infty$ as well as identifying $d_2, d_3$.

$\square$

We are now ready to derive Theorem 1.

**Theorem 1.** *For almost all CIDs $M = (G, P)$ satisfying Assumptions 1 and 2, we can identify the directed acyclic graph $G$ and joint distribution $P$ over all ancestors of the utility $\boldsymbol{Anc}_U$ given $\{\pi_\sigma^*(d \mid \boldsymbol{pa}_D)\}_{\sigma \in \Sigma}$ where $\pi_\sigma^*(d \mid \boldsymbol{pa}_D)$ is an optimal policy in the domain $\sigma$ and $\Sigma$ is the set of all mixtures of local interventions. Proof in Appendix C.*

*Proof.* We learn the graph $G$ and parameters $P(c_i \mid \mathbf{pa}_i)$ by learning 'leave-one-out' interventional distributions $P(c_i \mid \mathrm{do}(c_1, \ldots, c_{i-1}, c_{i+1}, \ldots, c_N))$. Note that under this intervention $C_i$ depends only on its parent set and hence $P(c_i \mid \mathrm{do}(c_1, \ldots, c_{i-1}, c_{i+1}, \ldots, c_N)) = P(c_i \mid \mathbf{pa}_i)$ where $\mathbf{Pa}_i = \mathbf{pa}_i$ denotes the state of $\mathbf{Pa}_i$ under the leave-one-out intervention. Almost all $P$ are causally faithful (Meek, 2013). Hence, for almost all $P$, these interventional distributions can be used to determine $\mathbf{Pa}_i$ as in the interventional distribution $C_i \not\perp C_j$ if and only if $C_j \in \mathbf{Pa}_i$. Explicitly, for almost all environments, $C_j \in \mathbf{Pa}_i$ if and only if there are two leave-one-out interventions that differ only on

---

**Algorithm 1** Identify $q_{\text{crit}}, d_2, d_3$ using policy oracle. Input: $(U, \Pi_{\Sigma}^*, N, \sigma)$

---

$d_1 \leftarrow \Pi_{\Sigma}^*(\sigma)$
$\sigma', d_2 \leftarrow$ any hard intervention on $\boldsymbol{C}$ s.t. $d_2 = \arg\max_d U(d, \boldsymbol{x}) \neq d_1$
$D(q = 1) \leftarrow \{d \text{ s.t. } d = \arg\max_d U(d, \boldsymbol{x}')\}$
$\theta = 0$
**for** $i \leftarrow 1$ to $N$ **do**
$\quad q \sim \text{Uniform}(0, 1)$
$\quad \pi^*(d \mid \mathbf{pa}_D) \leftarrow \Pi_{\Sigma}^*(\sigma_3(q))$
$\quad$**if** $d \in D(q = 1) \ \forall \pi^*(d \mid \mathbf{pa}_D) > 0$ **then**
$\quad\quad \theta \leftarrow \theta + 1$
$\quad$**end if**
**end for**
$q_{\text{crit}} = \theta/N$
$D(q_{\text{crit}}) \leftarrow \Pi_{\Sigma}^*(q_{\text{crit}}\sigma + (1 - q_{\text{crit}})\sigma')$
$d_3 \in D(q_{\text{crit}}), d_3 \neq d_2$
**return** $q_{\text{crit}}, d_2, d_3$

---

$C_j$ with $C_j = c_j$ and $C_j = c_j'$ such that $P(c_i \mid \text{do}(c_1, \ldots, c_j, \ldots, c_{i-1}, c_{i+1}, \ldots, c_N)) \neq P(c_i \mid \text{do}(c_1, \ldots, c_j', \ldots, c_{i-1}, c_{i+1}, \ldots, c_N))$. For ease of notation we will use $P(c_i \mid \mathbf{pa}_i)$ interchangeably with $P(c_i \mid \text{do}(c_1, \ldots, c_j, \ldots, c_{i-1}, c_{i+1}, \ldots, c_N))$.

First we learn the parameters for chance variables that have a directed path to $U$ that does not include $D$, i.e. are ancestors of $U$ in the graph $G_{\hat{D}}$ where we intervene on $D$.

**Case 1: learning parameters for $C_i \in \mathbf{Anc}_U(G_{\hat{D}})$.**

Consider a directed path $C_k \rightarrow \ldots \rightarrow C_1$ where $C_1 \in \mathbf{Pa}_U$ and all variables are chance nodes (the path does not include $D$). Assume we know $\mathbf{Pa}_{k-1}, \ldots, \mathbf{Pa}_1$ and the parameters $P(C_i \mid \mathbf{pa}_i)$ for $i = k-1, \ldots, 1$. We show that given these parameters we can identify the unknown parameters $P(c_k \mid \mathbf{pa}_k)$ (and hence $\mathbf{Pa}_k$). Define $\boldsymbol{Y} = \boldsymbol{C} \setminus \{C_k, \ldots, C_1\}$ and consider the local intervention $\sigma = \text{do}(y_1, \ldots, y_{N-k}, c_k = f(c_k))$ where $\text{do}(c_k = f(c_k))$ is a local intervention on $C_k$ such that,

$$f(C_k) = \begin{cases} c_k', & C_k = c_k' \\ c_k'' & \text{otherwise} \end{cases} \tag{18}$$

I.e. $f(C_k)$ maps $C_k$ to a 2 dimensional subspace where the image of $C_k = c_k'$ is $C_k = c_k'$ and all other states being mapped to $C_k = c_k''$, where $c_k', c_k'' \neq c_k'$ are arbitrary states of $C_k$. In the following we mask all inputs to the policy $\mathbf{Pa}_D' = \emptyset$.

By Lemma 4 we can identify,

$$\sum_{\boldsymbol{c}} P(\boldsymbol{C} = \boldsymbol{c}; \sigma)[U(d, \boldsymbol{c}) - U(d', \boldsymbol{c})]$$

$$= \sum_{c_k} \ldots \sum_{c_1} P(c_k \mid \mathbf{pa}_k; \sigma) \ldots P(c_1 \mid \mathbf{pa}_1; \sigma)[U(d, \boldsymbol{c}) - U(d', \boldsymbol{c})]$$

$$= \sum_{c_k} P(c_k \mid \mathbf{pa}_k; \sigma)\beta(c_k) \tag{19}$$

where

$$\beta(c_k) := \sum_{c_{k-1}} \ldots \sum_{c_1} P(c_{k-1} \mid \mathbf{pa}_{k-1}; \sigma) \ldots P(c_1 \mid \mathbf{pa}_1; \sigma)[U(d, \boldsymbol{c}) - U(d', \boldsymbol{c})] \tag{20}$$

and $\beta(c_1) := [U(d, \boldsymbol{c}) - U(d', \boldsymbol{c})]$. Note that in equation 19 $\beta(c_k)$ is determined by the known parameters $P(c_{k-1} \mid \mathbf{pa}_{k-1}), \ldots, P(c_1 \mid \mathbf{pa}_1)$ and $U(\mathbf{pa}_U)$, and $\beta(c_k)$ are non-zero for almost all $P, U$ as $\beta(c_k) = 0$ is a polynomial equation in these parameters it is not satisfied for almost all $P, U$.

Using the definition of the local intervention in equation 18 we have $P(C_k = c_k' \mid \mathbf{pa}_k; \sigma) = P(C_k = c_k' \mid \mathbf{pa}_k)$, and $P(C_k = c_k'' \mid \mathbf{pa}_k; \sigma) = 1 - P(C_i = c_k' \mid \mathbf{pa}_k; \sigma) = 1 - P(C_k = c_k' \mid \mathbf{pa}_k)$. Therefore the right hand side of equation 19 has a single undetermined parameter $P(C_k = c_k' \mid \mathbf{pa}_k)$

and the left hand side can be determined using the policy oracle (Lemma 4), and we can solve for $P(C_k = c'_k \mid \mathbf{pa}_k)$. By repeating this procedure with different interventions, varying the hard intervention $\mathrm{do}(\boldsymbol{Y} = \boldsymbol{y})$ and the choices of $c'_k, c''_k$, we can identify $P(c_k \mid \mathbf{pa}_k)$ for all $c_k, \mathbf{pa}_k$ and hence $\mathbf{Pa}_k$.

We now learn the parameters for all $C_i \in \mathbf{Anc}_U(G_{\hat{D}})$. We know the set $\mathbf{Pa}_U$ as this is the domain of the utility function $U(\mathbf{Pa}_U)$ which is known by assumption. We can then proceed iteratively, first learning the parameters of $P, G$ that are $P(c_1 \mid \mathbf{pa}_1)$ and $\mathbf{Pa}_1$ for some $C_1 \in \mathbf{Pa}_U$. We can do this as $\beta(c_1) = U(d, \boldsymbol{x}) - U(d', \boldsymbol{x})$ with $d, d'\boldsymbol{x}$ returned by Algorithm 1 in Lemma 4 and $U(\mathbf{Pa}_U)$ is known. We can then determine the parameters for all $C_j \in \mathbf{Pa}_1$, and so on until we have traversed $\mathbf{Anc}_1$. We repeat this for all $C_i \in \mathbf{Pa}_U$ until we have covered all $\mathbf{Anc}_U(G_{\hat{D}})$.

**Case 2: learning parameters for $C_i \in \mathbf{Anc}_D, C_i \notin \mathbf{Anc}_U(G_{\hat{D}})$.**

Consider $C_k \in \mathbf{Anc}_U$ for which all directed paths to $U$ are via $D$, $C_k \to C_{k-1} \to \ldots \to C_1$ where $C_1 \in \tilde{P}a_D$. As before, assume we know $\mathbf{Pa}_{k-1}, \ldots, \mathbf{Pa}_1$ and the parameters $P(C_i \mid \mathbf{pa}_i)$ for $i = k-1, \ldots, 1$. We now show that given these parameters we can identify the unknown parameters $P(c_k \mid \mathbf{pa}_k)$ (and hence $\mathbf{Pa}_k$). Define $\boldsymbol{Y} = \boldsymbol{C} \setminus \{C_k, \ldots, C_1\}$ and let $\sigma = \mathrm{do}(y_1, \ldots, y_{N-k}, c_k = f(c_k))$ where $\mathrm{do}(c_k = f(c_k))$ is a local intervention defined in equation 18. We now mask all evidence except $C_1$, i.e. $\mathbf{Pa}'_D = \{C_1\}$. Note that as $C_1 \notin \mathbf{Pa}_U$ we can apply Lemma 4, giving (for $k \geq 2$)

$$\sum_{\boldsymbol{z}} P(\boldsymbol{C} = \boldsymbol{c}; \sigma)[U(d, \boldsymbol{c}) - U(d', \boldsymbol{c})]$$

$$= \sum_{c_k} \ldots \sum_{c_2} P(c_k \mid \mathbf{pa}_k; \sigma) \ldots P(c_1 \mid \mathbf{pa}_1)[U(d, \boldsymbol{c}) - U(d', \boldsymbol{c})] \tag{21}$$

$$= \sum_{c_k} P(c_k \mid \mathbf{pa}_k)\alpha(c_k) \tag{22}$$

where $\boldsymbol{Z} = \boldsymbol{C} \setminus \{C_1\}$ and,

$$\alpha(c_k) := \sum_{c_{k-1}} \ldots \sum_{c_2} P(c_{k-1} \mid \mathbf{pa}_{k-1}) \ldots P(c_1 \mid \mathbf{pa}_1)[U(d, \boldsymbol{c}) - U(d', \boldsymbol{c})] \tag{23}$$

and for $k = 1$ we have,

$$\sum_{\boldsymbol{z}} P(\boldsymbol{C} = \boldsymbol{c}; \sigma)[U(d, \boldsymbol{c}) - U(d', \boldsymbol{c})] = P(C_1 = c_1 \mid \mathbf{pa}_1; \sigma)\alpha(1) \tag{24}$$

where $\alpha(1) := [U(d, \boldsymbol{x}) - U(d', \boldsymbol{x})]$. We can determine $\alpha(c_k)$ as we know the parameters for $C_{k-1}, \ldots, C_1$ by assumption, and $\alpha(c_k) \neq 0$ for almost all $P, U$ as the equation $\alpha(c_k) = 0$ is a polynomial in the model parameters by Lemma 2 it is not satisfied for almost all $P, U$. Using the definition of the local intervention equation 18 we have $P(C_k = c'_k \mid \mathbf{pa}_k; \sigma) = P(C_k = c'_k \mid \mathbf{pa}_k)$, and $P(C_k = c''_k \mid \mathbf{pa}_k; \sigma) = 1 - P(C_i = c'_k \mid \mathbf{pa}_k; \sigma) = 1 - P(C_k = c'_k \mid \mathbf{pa}_k)$. Therefore the right hand side of equation 19 has a single undetermined parameter $P(C_k = c'_k \mid \mathbf{pa}_k)$ and the left hand side can be determined using the policy oracle (using Lemma 4, noting $\mathbf{Pa}'_D = \{C_1\}$ and $\{C_1\} \cap \mathbf{Pa}_U = \emptyset$), and we can solve for $P(C_k = c'_k \mid \mathbf{pa}_k)$. By repeating this procedure with different interventions, varying the hard intervention $\mathrm{do}(\boldsymbol{Y} = \boldsymbol{y})$ and the choices of $c'_k, c''_k$, we can identify $P(c_k \mid \mathbf{pa}_k)$ for all $c_k, \mathbf{pa}_k$ and hence $\mathbf{Pa}_k$.

We now learn the parameters for all $C_i \in \mathbf{Anc}_D \setminus \mathbf{Anc}_U(G_{\hat{D}})$. We know $\mathbf{Pa}_D$ from the domain of the policy returned by the policy oracle. If the parameters for all variables in $\mathbf{Pa}_D$ have be learned in the previous set, we are finished. Otherwise, there are variables that are in $\mathbf{Anc}_U$ for which all directed paths to $U$ are via $D$. Let this set of variables by $\tilde{\mathbf{Pa}}_D \subseteq \mathbf{Pa}_D$. For any $C_1 \in \tilde{\mathbf{Pa}}_D$ we can determine $\alpha(c_1) = U(d, \boldsymbol{x}) - U(d', \boldsymbol{x})$ with $d, d', \boldsymbol{x}$ returned by Algorithm 1 in Lemma 4, noting that $C_1 \notin \mathbf{Pa}_U$. We can then determine the parameters for all $C_j \in \mathbf{Pa}_1$, and so on until we have traversed $\mathbf{Anc}_1$, and repeat until we have learned the parameters for all $C_i \in \mathbf{Anc}_D \setminus \mathbf{Anc}_U(G_{\hat{D}})$.

$\square$

# D   PROOF OF THEOREM 2

In this section we derive a version of Lemma 4 using a $\delta$-optimal policy oracle for $\delta > 0$. The reason we consider this case is that Theorem 1 assumes optimality, which is a strong assumption that won't be satisfied by realistic systems. It is therefore important to determine if our main results are contingent on this assumption. For example, it may be that we can only identify a causal model from the agent's policy for $\delta = 0$, and for $\delta > 0$ no causal model can be learned. Instead, what we find is that realistic agents with $\delta > 0$ have to learn approximate causal models, with the fidelity of these approximations increasing in a reasonable way as $\delta \to 0$.

**Low-regret analysis.** What is a reasonable way for the approximation errors to change with $\delta$? Clearly, if an agent has an arbitrarily large regret bound we cannot expect to learn anything about the environment from its policy. For example, a completely random policy can satisfy a large enough regret bound, and an agent does not need to learn anything about the environment to learn this policy. Therefore we must still constrain the regret to be small in our analysis, and the standard way to do this by an order analysis.

We define 'small regret' as $\delta \ll \mathbb{E}^{\pi^*}[U]$. As we work with the normalised utility function (see Appendix A.1), we have $\mathbb{E}^{\pi^*}[U] \leq 1$ and so we can define the small regret regime as $\delta \ll 1$. What we find is that for small $\delta$ the order of the error in our estimation of the model parameters grows linearly with the order of increase in the regret for agents that incur only a small regret. Therefore we get a linear trade-off between regret and accuracy for small $\delta$.

First we show that Algorithm 1 allows us to estimate the value of $Q = \sum_{\boldsymbol{z}} P(\boldsymbol{C} = \boldsymbol{c}; \sigma)[U(d, \boldsymbol{c}) - U(d', \boldsymbol{c})]$ with an approximate value $\tilde{Q}$, and estimate bounds $\tilde{Q}^{\pm}$ such that the true value of $Q$ is guaranteed to satisfy $\tilde{Q}^- \leq Q \leq \tilde{Q}^+$.

**Lemma 5.** *Using a $\delta$-optimal policy oracle $\Pi_{\Sigma}^{\delta}$ where $\Sigma$ includes all mixtures of local interventions, including masking inputs $\boldsymbol{Pa}'_D \subseteq \boldsymbol{Pa}_D$, then for any given $\boldsymbol{Pa}'_D = \boldsymbol{pa}'_D$ such that $\boldsymbol{Pa}'_D \cap \boldsymbol{Pa}_U = \emptyset$, we can determine $d, d', \boldsymbol{x}'$ where $d \neq d'$ and a point estimate $\tilde{Q}$ for $Q(\boldsymbol{pa}_D, d, d') := \sum_{\boldsymbol{z}} P(\boldsymbol{C} = \boldsymbol{c}; \sigma)[U(d, \boldsymbol{x}) - U(d', \boldsymbol{x})] < 0$ and bounds $Q \in [\tilde{Q}^-, \tilde{Q}^+]$ where $\boldsymbol{Z} = \boldsymbol{C} \setminus \boldsymbol{Pa}_D$, $\boldsymbol{X} = \boldsymbol{Pa}_U \setminus \{D\}$ and,*

$$\frac{1}{1-\xi}(Q - \delta) \leq \tilde{Q} \leq \frac{1}{1+\xi}(Q + \delta) \tag{25}$$

*where*

$$\xi := \delta/(U(d, \boldsymbol{x}') - U(d', \boldsymbol{x}')) > 0 \tag{26}$$

*and in the worst case these bounds scale with $\delta$ as*

$$\tilde{Q}^+ \leq \left(\frac{1-\xi}{1+\xi}\right)Q + \frac{2\delta}{1+\xi} \tag{27}$$

$$\tilde{Q}^- \geq \left(\frac{1+\xi}{1-\xi}\right)Q - \frac{2\delta}{1-\xi} \tag{28}$$

*Proof.* By Lemma 3 for almost all $P, U$ there is a single optimal decision following the shift $\sigma$. Let $d_1$ be the optimal decision returned by the policy oracle in the context $\boldsymbol{Pa}'_D = \boldsymbol{pa}'_D$, which must satisfy the bound $\mathbb{E}[u \mid d, \boldsymbol{pa}_D; \sigma] \leq \max_d \mathbb{E}[u \mid d, \boldsymbol{pa}_D; \sigma] - \delta$.

Consider a hard intervention on all $C_i \in \boldsymbol{C}$, $\sigma' := \mathrm{do}(c'_1, c'_2, \ldots, c'_N)$ where for all $C_i \in \boldsymbol{Pa}'_D$ we set $C_i = c_i$ to be the same state as in observation $\boldsymbol{Pa}'_D = \boldsymbol{pa}'_D$. The expected utility under this intervention is $\mathbb{E}[u \mid \mathrm{do}(D = d), \boldsymbol{pa}'_D; \sigma'] = U(d, \boldsymbol{x}')$ where $\boldsymbol{X} = \boldsymbol{Pa}_U \setminus \{D\}$ (and we have that $D \in \boldsymbol{Pa}_U$ from Lemma 1 iii)).

Next we show that there is a choice of hard intervention $\sigma'$ such that the policy oracle must return different optimal decisions in the context $\boldsymbol{Pa}'_D = \boldsymbol{pa}'_D$ for $\sigma'$ and $\sigma$. As $\boldsymbol{Pa}'_D \cap \boldsymbol{Pa}_U = \emptyset$ then we are free to choose any $X = x'$ and the resulting $\sigma'$ will be compatible with the evidence $\boldsymbol{Pa}'_D = \boldsymbol{pa}'_D$. Note that by Lemma 1 i) $\exists \, \boldsymbol{X} = \boldsymbol{x}'$ s.t. $d_1 \neq \arg\max_d U(d, x')$, else $D = d_1$ is optimal for all $\boldsymbol{X} = \boldsymbol{x}$ which violates domain dependence. We can determine this $\boldsymbol{X} = \boldsymbol{x}'$ given the utility function and $d_1$. Let $d_2 = \arg\max_d U(d, \boldsymbol{x}')$ and $\sigma' = \mathrm{do}(c'_1, c'_2, \ldots, c'_N)$ be the hard intervention for which $\boldsymbol{X} = \boldsymbol{x}'$ and $\boldsymbol{Pa}'_D = \boldsymbol{pa}'_D$. Note, we do not use the policy oracle to determine $d_2$ which can be

determined from $U(\mathbf{Pa}_U)$ alone, and hence there is no uncertainty if $d_2$ is in fact optimal under $\sigma'$ for any regret bound, nor that $d_1$ is not optimal under $\sigma'$.

Consider the joint distribution over $\boldsymbol{C}$ under the mixed local intervention $\tilde{\sigma}(q) = q\sigma + (1-q)\sigma'$,

$$P(\boldsymbol{C} = \boldsymbol{c} \mid \text{do}(D = d); \tilde{\sigma}(q)) = P(\boldsymbol{C} = \boldsymbol{c}; \tilde{\sigma}(q)) \tag{29}$$

$$= qP(\boldsymbol{C} = \boldsymbol{c}; \sigma) + (1-q)P(\boldsymbol{C} = \boldsymbol{c}; \sigma') \tag{30}$$

where in the first line we have used $\mathbf{Ch}_D = \{U\}$ to drop the intervention. Note that $\boldsymbol{Z} = \boldsymbol{C}\backslash\mathbf{Pa}_D \neq \emptyset$ by Lemma 1 i). The expected utility is given by,

$$\mathbb{E}[u \mid \mathbf{pa}_D, \text{do}(D = d); \tilde{\sigma}(q)] = \sum_{\boldsymbol{z}} P(\boldsymbol{Z} = \boldsymbol{z} \mid \mathbf{pa}_D, \text{do}(D = d); \tilde{\sigma}(q))U(d, \boldsymbol{x}) \tag{31}$$

$$= \sum_{\boldsymbol{z}} \frac{P(\boldsymbol{C} = \boldsymbol{c} \mid \text{do}(D = d); \tilde{\sigma}(q))}{P(\mathbf{pa}_D \mid \text{do}(D = d); \tilde{\sigma}(q))}U(d, \boldsymbol{x}) \tag{32}$$

$$= \frac{1}{P(\mathbf{pa}_D; \tilde{\sigma}(q))} \sum_{\boldsymbol{z}} P(\boldsymbol{C} = \boldsymbol{c}; \tilde{\sigma}(q))U(d, \boldsymbol{x}) \tag{33}$$

$$= \frac{1}{P(\mathbf{pa}_D; \tilde{\sigma}(q))} \sum_{\boldsymbol{z}} qP(\boldsymbol{C} = \boldsymbol{c}; \sigma)U(d, \boldsymbol{x}) + (1-q)P(\boldsymbol{C} = \boldsymbol{c}; \sigma')U(d, \boldsymbol{x}') \tag{34}$$

Note that for $q = 1$ the optimal decision is $d_1$ and for $q = 0$ the optimal decision returned by the policy oracle belongs to the set $\{d \text{ s.t. } d = \arg\max_d U(d, \boldsymbol{x}')\}$ which does not contain $d_1$. Furthermore, the argmax of equation 34 with respect to $d$ is a piecewise linear function with domain $q \in [0, 1]$. Therefore there must be some $q = q_{\text{crit}}$ that is the smallest value of $q$ such that for $q < q_{\text{crit}}$ the policy oracle returns an optimal decision in the set $\{d \text{ s.t. } d = \arg\max_d U(d, \boldsymbol{x}')\}$ and for $q \geq q_{\text{crit}}$ the optimal decision is not in this set. The value of $q_{\text{crit}}$ is given by $\mathbb{E}[u \mid \mathbf{pa}_D, \text{do}(D = d); \tilde{\sigma}(q_{\text{crit}})] = 0$, which by equation 34 is,

$$q_{\text{crit}} \sum_{\boldsymbol{z}} P(\boldsymbol{C} = \boldsymbol{c}; \sigma)[U(d_2, \boldsymbol{x}) - U(d_3, \boldsymbol{x})] + (1 - q_{\text{crit}})[U(d_2, \boldsymbol{x}') - U(d_3, \boldsymbol{x}')] = 0 \tag{35}$$

where $d_2 \in \{d \text{ s.t. } d = \arg\max_d U(d, \boldsymbol{x}')\}$ and $d_3 \notin \{d \text{ s.t. } d = \arg\max_d U(d, \boldsymbol{x}')\}$. This yields the following expression for $q_{\text{crit}}$,

$$q_{\text{crit}} = \left(1 - \frac{\sum_{\boldsymbol{z}} P(\boldsymbol{C} = \boldsymbol{c}; \sigma)[U(d_2, \boldsymbol{x}) - U(d_3, \boldsymbol{x})]}{U(d_2, \boldsymbol{x}') - U(d_3, \boldsymbol{x}')}\right)^{-1} \tag{36}$$

where we have used $\sum_{\boldsymbol{z}} P(\boldsymbol{C} = \boldsymbol{c}; \sigma')[U(d_2, \boldsymbol{c}) - U(d_3, \boldsymbol{c})] = U(d_2, \boldsymbol{x}') - U(d_3, \boldsymbol{x}')$.

While Algorithm 1 identifies the smallest value of $q$ such that the optimal policy changes, as we no longer have an optimal policy oracle, the probability $\tilde{q}$ returned by Algorithm 1 is no longer necessarily equal to $q_{\text{crit}}$. Instead, there are minimal and maximal value of $\tilde{q}$ that Algorithm 1 can return, which are determined by the regret bound (see Figure 7).

Our first aim is to bound $q_{\text{crit}}$ using $\tilde{q}$ returned by the policy oracle. The maximal (minimal) values $\tilde{q}$ can take while satisfying the regret bound are $q^{\pm}$, which are the solutions to the equations

$$\delta \geq \mathbb{E}[u \mid \mathbf{pa}_D, \text{do}(D = d_2); \tilde{\sigma}(q^+)] - \mathbb{E}[u \mid \mathbf{pa}_D, \text{do}(D = d_3); \tilde{\sigma}(q^+)] \tag{37}$$

$$-\delta \leq \mathbb{E}[u \mid \mathbf{pa}_D, \text{do}(D = d_2); \tilde{\sigma}(q^-)] - \mathbb{E}[u \mid \mathbf{pa}_D, \text{do}(D = d_3); \tilde{\sigma}(q^-)] \tag{38}$$

Using equation 34 these simplify to

$$\delta P(\mathbf{pa}_D; \sigma(q^+)) \geq q^+ \sum_{\boldsymbol{z}} P(\boldsymbol{C} = \boldsymbol{c}; \sigma)[U(d_2, \boldsymbol{x}) - U(d_3, \boldsymbol{x})] + (1 - q^+)[U(d_2, \boldsymbol{x}') - U(d_3, \boldsymbol{x}')]$$

$$\tag{39}$$

$$\delta P(\mathbf{pa}_D; \sigma(q^-)) \leq q^- \sum_{\boldsymbol{z}} P(\boldsymbol{C} = \boldsymbol{c}; \sigma)[U(d_2, \boldsymbol{x}) - U(d_3, \boldsymbol{x})] + (1 - q^-)[U(d_2, \boldsymbol{x}') - U(d_3, \boldsymbol{x}')]$$

$$\tag{40}$$

We can relax and simplify these bounds by taking the maximum possible values for the unknown quantity $P(\mathbf{pa}_D; \sigma(\tilde{q})) \to 1$ giving,

$$\delta \geq q^+ \sum_{z} P(\boldsymbol{C} = \boldsymbol{c}; \sigma)[U(d_2, \boldsymbol{x}) - U(d_3, \boldsymbol{x})] + (1 - q^+)[U(d_2, \boldsymbol{x}') - U(d_3, \boldsymbol{x}')] \qquad (41)$$

$$\delta \leq q^- \sum_{z} P(\boldsymbol{C} = \boldsymbol{c}; \sigma)[U(d_2, \boldsymbol{x}) - U(d_3, \boldsymbol{x})] + (1 - q^-)[U(d_2, \boldsymbol{x}') - U(d_3, \boldsymbol{x}')] \qquad (42)$$

Let $\Delta_1 := \mathbb{E}[u \mid \mathbf{pa}_D, \mathrm{do}(D = d_2); \sigma] - \mathbb{E}[u \mid \mathbf{pa}_D, \mathrm{do}(D = d_3); \sigma]$ and $\Delta_0 := U(d_2, \boldsymbol{x}') - U(d_3, \boldsymbol{x}')$. Note that $\Delta_0 > 0$ as $d_2$ is optimal under $\sigma'$, and $\Delta_1 < 0$ as by linearity we have $\mathbb{E}[u \mid \mathbf{pa}_D, \mathrm{do}(D = d_3); \tilde{\sigma}(q)] > \mathbb{E}[u \mid \mathbf{pa}_D, \mathrm{do}(D = d_2); \tilde{\sigma}(q)]$ for $q > q_{\mathrm{crit}}$ and $q_{\mathrm{crit}} < 1$ therefore $\mathbb{E}[u \mid \mathbf{pa}_D, \mathrm{do}(D = d_3); \tilde{\sigma}(1)] > \mathbb{E}[u \mid \mathbf{pa}_D, \mathrm{do}(D = d_2); \tilde{\sigma}(1)]$. We now define $q^{\pm}$ w.r.t the (relaxed) bounds equation 41 and equation 42, and simplifying these inequalities using equation 36 gives

$$\tilde{q} \leq q_+ = \min\{1, q_{\mathrm{crit}}(1 + \xi)\} \qquad (43)$$
$$\tilde{q} \geq q_- = \max\{0, q_{\mathrm{crit}}(1 - \xi)\} \qquad (44)$$

where

$$\xi := \delta/\Delta_0 > 0 \qquad (45)$$

We therefore generate bounds on $q_{\mathrm{crit}}$ using equation 44 and equation 43, i.e.

$$q_{\mathrm{crit}} \leq \tilde{q}/(1 - \xi) \qquad (46)$$
$$q_{\mathrm{crit}} \geq \tilde{q}/(1 + \xi) \qquad (47)$$

We use $\tilde{q}$ in place of $q_{\mathrm{crit}}$ in equation 36, giving an estimate $\tilde{Q}$ for $Q = \sum_{z} P(\boldsymbol{C} = \boldsymbol{c}; \sigma)[U(d_2, \boldsymbol{x}) - U(d_3, \boldsymbol{x})]$, yielding,

$$\tilde{Q} = \Delta_0 \left(1 - 1/\tilde{q}\right) \qquad (48)$$

Finally, applying bounds equation 46 and equation 47 gives,

$$\frac{1}{1 - \xi} \left(\Delta_0 - \delta\right) \leq \tilde{Q} \leq \frac{1}{1 + \xi} \left(\Delta_0 + \delta\right) \qquad (49)$$

Next, we determine upper and lower bounds $Q^{\pm}(\tilde{q})$ using $Q = \Delta_0(1 - 1/q_{\mathrm{crit}})$ and equation 46 and equation 47 giving,

$$Q \leq \tilde{Q}^+(\tilde{q}) = \Delta_0 \left(1 - \frac{1 - \xi}{\tilde{q}}\right) \qquad (50)$$

$$Q \geq \tilde{Q}^-(\tilde{q}) = \Delta_0 \left(1 - \frac{1 + \xi}{\tilde{q}}\right) \qquad (51)$$

noting that as equation 48 is monotonic in $q$ that the true value of $Q$ is guaranteed to fall between these bounds. Finally, we derive expressions for the worst-case bounds in terms of the true value of $Q$, which are given by determining $\tilde{Q}^{\pm}$ for the max and min values of $\tilde{q}$ which are given by equation 47 and equation 46,

$$Q^+ = \max_{\tilde{q}} \tilde{Q}^+(\tilde{q}) = \Delta_0 \left(1 - \frac{1 - \xi}{1 + \xi}\frac{1}{q_{\mathrm{crit}}}\right) \qquad (52)$$

$$= \left(\frac{1 - \xi}{1 + \xi}\right) Q + \frac{2\delta}{1 + \xi} \qquad (53)$$

$$Q^- = \min_{\tilde{q}} \tilde{Q}^-(\tilde{q}) = \Delta_0 \left(1 - \frac{1 + \xi}{1 - \xi}\frac{1}{q_{\mathrm{crit}}}\right) \qquad (54)$$

$$= \left(\frac{1 + \xi}{1 - \xi}\right) Q - \frac{2\delta}{1 - \xi} \qquad (55)$$

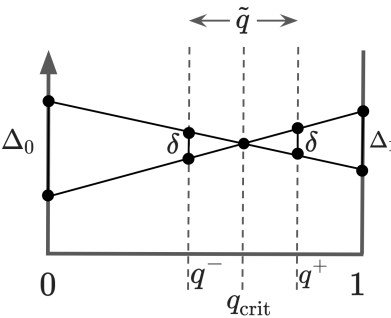

Figure 7: Overview of Lemma 5. $\Delta_0 = U(d_2, \boldsymbol{x}') - U(d_3, \boldsymbol{x}')$ and $\Delta_1 = \mathbb{E}[u \mid \mathbf{pa}_D, \mathrm{do}(D = d_2); \sigma] - \mathbb{E}[u \mid \mathbf{pa}_D, \mathrm{do}(D = d_3); \sigma]$. Using an optimal policy oracle we can identify $q_{\mathrm{crit}}$ precisely as detailed in Lemma 4. For $\delta > 0$ instead of returning $q_{\mathrm{crit}}$ Algorithm 1 returns $\tilde{q}$, as the agent can incur regret and so the value of $q$ for which the policy changes is no longer constrained to be $q_{\mathrm{crit}}$. We use $\tilde{q}$ in place to $q_{\mathrm{crit}}$ to calculate an approximate value of the target query, in the same way as in Lemma 4. The maximum and minimum values of $\tilde{q}$ can take are $q^{\pm}$ which result in maximal regret $\delta$, $\tilde{q} \geq q^- = q_{\mathrm{crit}}(1 - \delta/\Delta_0)$ and $\tilde{q} \leq q^+ = q_{\mathrm{crit}}(1 + \delta/\Delta_0)$. We can therefore bound the amount that $\tilde{Q}$ deviates from the value of the target query $Q$.

$\square$

**Lemma 6.** *For $\delta \ll \mathbb{E}^{\pi^*}[U]$, $\tilde{Q}$ and $\tilde{Q}^{\pm}$ (as defined in Lemma 5) satisfy bounds,*

$$\left|\tilde{Q} - Q\right| \leq \delta(1 - \frac{Q}{\Delta_0}) + \mathcal{O}(\delta^2) \tag{56}$$

*and*

$$\tilde{Q}^+ - Q \leq 2\delta(1 - \frac{Q}{\Delta_0}) + \mathcal{O}(\delta^2) \tag{57}$$

$$Q - \tilde{Q}^- \geq -2\delta(1 - \frac{Q}{\Delta_0}) + \mathcal{O}(\delta^2) \tag{58}$$

*Proof.* As we work with the normalised utility function (see Appendix A.1), we have $\mathbb{E}^{\pi^*}[U] \leq 1$ and so we can define the small regret regime as $\delta \ll 1$. We can Taylor expand the bounds on $\tilde{Q}, \tilde{Q}^{\pm}$ about $\delta = 0$ giving,

$$Q - \delta(1 - \frac{Q}{\Delta_0}) + \mathcal{O}(\delta^2) \leq \tilde{Q} \leq Q + \delta(1 - \frac{Q}{\Delta_0}) + \mathcal{O}(\delta^2) \tag{59}$$

and therefore,

$$\left|\tilde{Q} - Q\right| \leq \delta(1 - \frac{Q}{\Delta_0}) + \mathcal{O}(\delta^2) \tag{60}$$

and

$$\tilde{Q}^+ - Q \leq 2\delta(1 - \frac{Q}{\Delta_0}) + \mathcal{O}(\delta^2) \tag{61}$$

$$Q - \tilde{Q}^- \geq -2\delta(1 - \frac{Q}{\Delta_0}) + \mathcal{O}(\delta^2) \tag{62}$$

therefore for small $\delta$ the worst case error on our estimate $\tilde{Q}$ grows linearly in $\delta$, and our upper and lower bounds for $\tilde{Q}$ also grow linearly. $\square$

**Theorem 2.** *For almost all CIDs $M = (G, P)$ satisfying Assumptions 1 and 2, we can identify an approximate causal model $M' = (P', G')$ given $\{\pi_\sigma(d \mid \boldsymbol{pa}_D)\}_{\sigma \in \Sigma}$ where $\mathbb{E}^{\pi_\sigma}[U] \geq \mathbb{E}^{\pi^*_\sigma}[U] - \delta$ and $\Sigma$ is the set of mixtures of local interventions. The parameters of $M'$ satisfy $|P'(v_i \mid \boldsymbol{pa}_i) - P(v_i \mid \boldsymbol{pa}_i)| \leq \gamma(\delta) \; \forall \; V_i \in \boldsymbol{V}$ where $\gamma(0) = 0$ and $\gamma(\delta)$ grows linearly in $\delta$ for small regret $\delta \ll \mathbb{E}^{\pi^*}[U]$. Proof in Appendix D.*

*Proof.* We use the $\delta-$optimal policy oracle to estimate the model parameters following the same steps as in the proof of Theorem 1 in Appendix C. However, as the policy oracle is no longer optimal, the parameters estimates will have errors. Here, we show that for the parameters of $P$ these errors grow linearly in $\delta$ for $\delta \ll 1$, and that we learn a sparse sub-graph of $G$.

**Estimating parameters of $P$.**

In the proof of Theorem 1 we estimate the parameters $P(c_i \mid \mathbf{pa}_i)$ in two cases.

Case 1.

$$Q_k = \sum_{\mathbf{c}} P(\mathbf{C} = \mathbf{c}; \sigma)[U(d, \mathbf{c}) - U(d', \mathbf{c})] = \sum_{c_k} P(c_k \mid \mathbf{pa}_k; \sigma)\beta(c_k) \tag{63}$$

where

$$\beta(c_k) := \sum_{c_{k-1}} \ldots \sum_{c_1} P(c_{k-1} \mid \mathbf{pa}_{k-1}; \sigma) \ldots P(c_1 \mid \mathbf{pa}_1; \sigma)[U(d, \mathbf{c}) - U(d', \mathbf{c})] \tag{64}$$

which we rearrange using $P(c'_k \mid \mathbf{pa}_k; \sigma) = 1 - P(c''_k \mid \mathbf{pa}_k; \sigma)$ to give,

$$P(c'_k \mid \mathbf{pa}_k; \sigma) = \frac{Q_k - \beta(c''_k)}{\beta(c'_k) - \beta(c''_k)} \tag{65}$$

Assume we have approximate values $\hat{P}(c'_{k-1} \mid \mathbf{pa}_{k-1}; \sigma), \ldots, \hat{P}(c'_1 \mid \mathbf{pa}_1; \sigma)$ where $\hat{P}(c'_k \mid \mathbf{pa}_k; \sigma) = P(c'_k \mid \mathbf{pa}_k; \sigma) + \mathcal{O}(\delta)$, i.e. errors in our estimates for these parameters grow linearly in $\delta$ for $\delta \ll 1$. As $\beta(c_k)$ is a sum of products of these parameter estimates, then our estimate of $\beta(c_k)$ also has linear error for $\delta \ll 1$, i.e. $\hat{\beta}(c_k) = \beta(c_k) + \mathcal{O}(\delta)$, and likewise,

$$\hat{P}(c'_k \mid \mathbf{pa}_k; \sigma) = \frac{Q_k - \beta(c''_k) + \mathcal{O}(\delta)}{\beta(c'_k) - \beta(c''_k) + \mathcal{O}(\delta)} = P(c'_k \mid \mathbf{pa}_k; \sigma)(1 + \mathcal{O}(\delta)) \tag{66}$$

Then for $k = 1$ we know $\beta(c_1) = U(d, \mathbf{c}) - U(d', \mathbf{c})$ precisely, and so

$$\hat{P}(c'_1 \mid \mathbf{pa}_1; \sigma) = \frac{Q_1 - \beta(c''_1) + \mathcal{O}(\delta)}{\beta(c'_1) - \beta(c''_1)} = P(c'_1 \mid \mathbf{pa}_1; \sigma)(1 + \mathcal{O}(\delta)) \tag{67}$$

Which satisfies our assumption of $\mathcal{O}(\delta)$ error for $k = 1$, $\delta \ll 1$. Therefore for all $k$ we have error that grows linearly in $\delta$ for $\delta \ll 1$.

The expressions for $Q_k, \alpha(c_k)$ for case 2 in the proof of Theorem 1 are similar, and it is trivial to show by the same method that for these parameters the error also grow linearly in $\delta$ for $\delta \ll 1$.

**Learning graph structure.** In Theorem 1 we determine $\mathbf{Pa}_k$ from $P(c_k \mid \text{do}(\mathbf{C} \setminus \{C_k\}))$. Assuming causal faithfulness, which is satisfied for almost all $P$ (Meek, 2013), $C_j \in \mathbf{Pa}_k$ if and only if $P(c_k \mid \text{do}(\mathbf{C} \setminus \{C_k\}))$ differ for some $C_j = c_j, C_j = c'_j$. However, as we now only have estimates $\hat{P}(c_k \mid \text{do}(\mathbf{C} \setminus \{C_k\}))$, any variation with respect to $C_j = c_j$ may be due to the varying errors in these estimates rather than variation in the conditional probability itself. However, we have shown that we can learn any $P(c_i \mid \mathbf{pa}_i)$ within error bounds, and that these bounds scale linearly with $\delta$ for $\delta \ll 1$. Let $C_j \in \mathbf{Pa}_{i+n}$ and $\theta_{kj} = P(c_k \mid \text{do}(\mathbf{C} \setminus \{C_k\}))$, and denote the corresponding upper and lower bounds from Lemma 6 as $\theta_{kj}^{\pm}$. If $\exists \theta_{kj} \neq \theta_{kj'}$ and either $\theta_{kj}^+ < \theta_{kj'}^-$ or $\theta_{kj'}^+ < \theta_{kj}^-$, non-overlapping bounds for $C_j = c_j$ and $C_j = c'_j$, then we know with certainty that $C_j \in \mathbf{Pa}_k$. If there are no such non-overlapping bounds for all $j$, we do not know if $C_j \in \mathbf{Pa}_k$ and so exclude it from the set. This approach is guaranteed to identify a sub-graph of $G$ (i.e. no false positives—directed edges present in the approximate CBN that are not present in the environment). Further, we only miss a parent if in the true underlying causal model for all $\mathbf{Pa}_k = \mathbf{pa}_k$ intervening to change $C_j$ gives $|P(c_k \mid \mathbf{pa}_k, \text{do}(c_j)) - P(c_k \mid \mathbf{pa}_k, \text{do}(c'_j))| < \mathcal{O}(\delta)$. Hence for $\delta \ll 1$ we only fail to learn causal relations that small in magnitude (with respected to the regret $\delta$), i.e. where the causal effect of the parent on the child is $\mathcal{O}(\delta)$.

In Appendix F we explore the relation between the regret bound and the error in the learned causal graph using simulated data, and find that even agents that incur relatively high regret can be used to identify causal structure to a high accuracy compared to a random baseline. □

## E    APPENDIX: PROOF OF THEOREM 3

**Theorem 3.** *Given the CBN $M = (P, G)$ that is causally sufficient we can identify optimal policies $\pi_\sigma^*(d \mid \boldsymbol{pa}_D)$ for any given $U$ where $\boldsymbol{Pa}_U \subseteq \boldsymbol{C}$ and for all soft interventions $\sigma$. Given an approximate causal model $M' = (P', G')$ for which $|P'(v_i \mid \boldsymbol{pa}_i) - P(v_i \mid \boldsymbol{pa}_i)| \leq \epsilon \ll 1$, we can identify regret-bounded policies where the regret $\delta$ grows linearly in $\epsilon$. Proof in Appendix E.*

*Proof.* First we consider the case where we know the exact model $M = (P, G)$. As $M$ is causally sufficient we can identify $\mathbb{E}[u \mid d, \mathbf{pa}_D; \sigma]$ for any given soft interventions compatible with $G$ and which involve only variables in $G$ (which includes $\mathbf{Anc}_U \cup \{U\}$). Our policy oracle is constructed by i) estimating $\mathbb{E}[u \mid d, \mathbf{pa}_D; \sigma]$ for the input $\sigma$, ii) calculating $d^* = \arg\max_d \mathbb{E}[u \mid d, \mathbf{pa}_D; \sigma]$ and returning any $d^*$ satisfying this.

Next, consider the case where we know the approximate model $M' = (P', G')$, for which $|P'(v_i \mid \mathbf{pa}_i) - P(v_i \mid \mathbf{pa}_i)| \leq \epsilon \ll 1$ which implies $P'(v_i \mid \mathbf{pa}_i) = P(v_i \mid \mathbf{pa}_i) + c_i \epsilon$ where $|c_i| \leq 1$. First we show that for any soft intervention $\sigma$ we can approximate the post-intervention joint distribution $P'(\boldsymbol{Z} = \boldsymbol{z} \mid \mathrm{do}(D = d), \mathbf{Pa}_D = \mathbf{pa}_D; \sigma) = P(\boldsymbol{Z} = \boldsymbol{z} \mid \mathrm{do}(D = d), \mathbf{Pa}_D = \mathbf{pa}_D; \sigma) + k\epsilon + \mathcal{O}(\epsilon^2)$ where $\boldsymbol{Z} = \boldsymbol{C} \setminus \mathbf{Pa}_D$ and $k$ is a function of the model parameters and constant in $\epsilon$. Let $\sigma = \sum_j q_j \sigma_j$ where $\sigma_j$ are soft interventions.

$$P'(\boldsymbol{Z} = \boldsymbol{z} \mid \mathrm{do}(D = d), \mathbf{Pa}_D = \mathbf{pa}_D; \sigma) = \sum_j q_j \frac{P'(\boldsymbol{C} = \boldsymbol{c} \mid \mathrm{do}(D = d); \sigma)}{P'(\boldsymbol{Z} = \boldsymbol{z}', \mathbf{Pa}_D = \mathbf{pa}_D; \sigma_j)} \tag{68}$$

$$= \sum_j q_j \frac{\prod_i P'(C_i = c_i \mid \mathrm{do}(D = d); \sigma_j)}{\sum_{\boldsymbol{z}'} \prod_i P'(C_i = c_i' \mid \mathrm{do}(D = d); \sigma_j)} \tag{69}$$

$$= \sum_j q_j \frac{\prod_i [P(C_i = c_i \mid \mathrm{do}(D = d); \sigma_j) + c_i\epsilon]}{\sum_{\boldsymbol{z}'} \prod_i [P(C_i = c_i' \mid \mathrm{do}(D = d); \sigma_j) + c_i\epsilon]} \tag{70}$$

$$= \sum_j q_j \frac{\prod_i P(C_i = c_i \mid \mathrm{do}(D = d); \sigma_j)(1 + c_{ij}'\epsilon)}{\sum_{\boldsymbol{z}'} \prod_i P(C_i = c_{ij}' \mid \mathrm{do}(D = d); \sigma_j)(1 + c_i'\epsilon)} \tag{71}$$

$$= P(\boldsymbol{Z} = \boldsymbol{z} \mid \mathrm{do}(D = d), \mathbf{Pa}_D = \mathbf{pa}_D; \sigma) + \epsilon f(\theta) + \mathcal{O}(\epsilon^2) \tag{72}$$

where $c_{ij}' := c_i / P(C_i = c_i \mid \mathrm{do}(D = d); \sigma_j)$ and $f(\theta)$ is a polynomial in the model parameters $\theta_i = P(v_i \mid \mathbf{pa}_i)$. Therefore the expected utility under intervention $\sigma$ evaluated using $M'$ satisfies,

$$\mathbb{E}_{P'}[U \mid \mathrm{do}(D = d), \mathbf{Pa}_D = \mathbf{pa}_D] = \sum_{\boldsymbol{z}} P'(\boldsymbol{Z} = \boldsymbol{z} \mid \mathrm{do}(D = d), \mathbf{Pa}_D = \mathbf{pa}_D; \sigma) \tag{73}$$

$$= \sum_{\boldsymbol{z}}{}' (\boldsymbol{Z} = \boldsymbol{z} \mid \mathrm{do}(D = d), \mathbf{Pa}_D = \mathbf{pa}_D; \sigma) + \epsilon g(\theta) + \mathcal{O}(\epsilon^2) \tag{74}$$

$$= \mathbb{E}[U \mid \mathrm{do}(D = d), \mathbf{Pa}_D = \mathbf{pa}_D] + \epsilon g(\theta) + \mathcal{O}(\epsilon^2) \tag{75}$$

where $g(\theta)$ is a polynomial in the model parameters. The decision $d^* = \arg\max_d \mathbb{E}_{P'}[U \mid \mathrm{do}(D = d), \mathbf{Pa}_D = \mathbf{pa}_D]$ incurs at most $\epsilon g(\theta)$ regret, and therefore the regret is linear in $\epsilon$. □

## F    EXPERIMENTS

As discussed in Section 4 the proofs of Theorems 1 and 2 can be viewed as causal discovery algorithms where we assume i) knowledge of the set of environment variables $\boldsymbol{C}$, ii) knowledge of the utility

function $U$, iii) the decision task is unmediated and iv) domain dependence. Given these assumptions we can learn an approximation of the underlying CBN given only the policy of the agent $\pi(\sigma)$ under interventions $\sigma$, with the approximation being exact when $\pi(\sigma)$ are optimal.

To demonstrate this theoretical result we take the proof for simple Binary decision tasks outlined in Appendix B and recast it as a causal discovery algorithm (Algorithm 2 below). We test it on CIDs of the form shown in Figure 6 where we randomly choose the joint distribution over $X, Y$ and their causal structure $G$. Note that Algorithm 2 is significantly simpler than the general method outlined in the proof of Theorem 1, as it exploits the fact that $D, X, Y$ are binary variables and that $|\boldsymbol{C}| = 2$. This causal discovery algorithm requires that we can intervene on the latent variables $X, Y$, but only requires that we can observe the response of a single variable (the decision) to these interventions. To motivate this setting, we can imagine situations where the latents $X, Y$ cannot be directly observed but can be intervened on.

**Example.** Many diseases cannot be directly observed in patient physiology, but can only be indirectly observed through the presence of symptoms. Let $X, Y \in \{0, 1\}$ be two such diseases, for which there are treatments, i.e. we can intervene to 'turn off' $X$ and $Y$ but cannot observe them. $D \in \{0, 1\}$ represents a decision to provide a specific pain relief medication, which results in a change in the symptom severity (utility). The response to pain relief depends on the presence or absence of the diseases (e.g pain relief is highly effective for patients with $X = T$, moderately effective for $Y = T$ and less effective for $X = F, Y = F$). The doctor's goal is to minimise symptom severity while avoiding unnecessary use of pain medication, e.g. $U(d, x, y) = d[s(x, y) - c]$ where $c$ is some cost associated with pain relief and $s(x, y)$ is the response to pain relief. Following an intervention $\sigma$ (e.g. curing a disease $\sigma = \mathrm{do}(X = F)$), the doctor adapts their treatment policy in the shifted population. For example, this adaptation could occur by trial and error, with the doctor choosing random treatment decisions $D$ and observing the change in symptom severity—a context-free bandit problem. Although we cannot directly observe the disease states $X, Y$, by intervening on the latent disease state and observing how the doctor's policy adapts, we can learn both the joint distribution $P(X, Y)$ and the causal graph over $X, Y$.

Figure 9 shows the average error in the learned parameters $P(x, y)$ and $G$ when $\pi(\sigma)$ satisfy different regret bounds. The results are averaged over 1000 randomly generated CBNs where i) the parameters of the joint distribution $P(x, y)$ are chosen at random, ii) the DAG $G$ over $X, Y$ is chosen at random from $X \rightarrow Y$ and $X \leftarrow Y$, iii) the utility function $U(d, x, y) \in [0, 1]$ is chosen at random (see Appendix A.2 for description of parameters). To simulate the regret-bounded agent we calculate the optimal policy for each environment and if the sub-optimal decision satisfies the regret bound we choose randomly from the two decisions when sampling from the policy oracle in Algorithm 1. We also compare to a random baseline algorithm which estimates $P(x, y) = 1/4$ and randomly selects from $X \rightarrow Y$ or $X \leftarrow Y$ with equal probability. In a small number of cases Algorithm 1 fails to predict $P(x, y) \in [0, 1]$ due to finite sample errors, and for these cases we replace the output of the causal discovery algorithm with a random guess.

From Figure 9 it appears that the error rate grows sub-linearly with regret. Note that the relevant scale for the regret is the difference in expected utility between the two decisions, hence we plot the normalised regret bound where we divide $\delta$ by this expected utility difference. Note that even for relatively large regret bounds, representing agents that generalise weakly, we can still identify the causal structure with a high accuracy. For example when the regret bound is 30% of the expected utility difference, we can still identify the correct causal structure in $\sim 90\%$ of the randomly generated CIDs. This describes an agent that is guaranteed to incur a regret of at most 30% of the expected utility difference between the decisions *before* the domain shift. If the domain shift results in the expected utility difference being less that 30% of the unshifted expected utility difference, the agent can return a sub-optimal decision.

## G  APPENDIX: TRANSPORTABILITY & PEARL'S CAUSAL HIERARCHY

**Transportability.** The problem of evaluating policies under distributional shifts has been studied extensively in transportability theory (Pearl & Bareinboim, 2011; Bareinboim & Pearl, 2016; Bellot & Bareinboim, 2022). For decision tasks as outlined in Section 2.2, transportability aims to provide necessary and sufficient conditions for identifying the expected utility following a distributional shift, $R = \mathbb{E}[u \mid d, \mathbf{pa}_D; \sigma]$, given (partial) knowledge of i) the joint $P$, causal graph $G$ and interventional

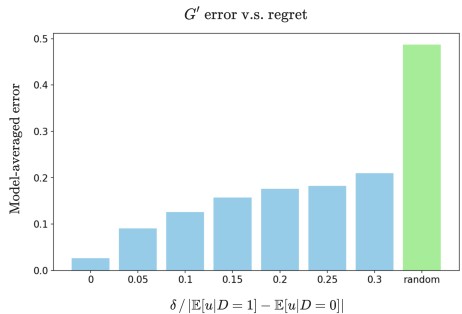

(a) Misclassification rate for G scaling with regret bound

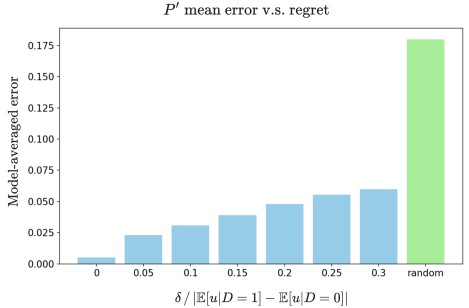

(b) Mean parameter error for P(x, y) scaling with regret bound

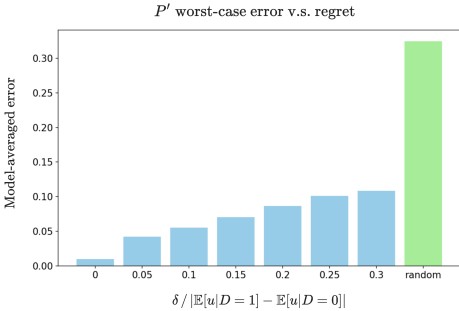

(c) Worst-case error for P(x, y) scaling with regret bound

Figure 9: Comparing the model-average error rates for a) the learned DAG and b) the mean error for parameters $P(x, y)$ and c) the worst-case error for parameters $P(x, y)$, v.s. the (normalised) regret bound $\delta / |\mathbb{E}[u \mid D = 1] - \mathbb{E}[u \mid D = 0]|$. Average error taken over 1000 randomly generated environments with binary decision $D$ and two binary latent variables $X, Y$. Comparison to error rate for random guess (green). Results appear to show sub-linear growth in error rate with regret bound. Note that even weakly generalising agents can be used to identify causal structure significantly better than the random baseline.

---

**Algorithm 2** Graph Learner for simple CID

---

1: **function** GRAPH LEARNER($\Pi_\Sigma^\delta$, $U$, $\delta$, $N$)
2:     $d_1, d_2, x', y', q_{\text{crit}} \leftarrow$ Algorithm 1$(U, \Pi_\Sigma^\delta, N, \sigma_1 = \text{do}(Y = 0))$        $\triangleright$ Identify $q_{\text{crit}}$ for do$(Y = 0)$
3:       Exp. U difference $= (U(d_2, x', y') - U(d_1, x', y')) * (1 - 1/q_{\text{crit}})$
4:       $\Delta_0 = U(0, 0, d_2) - U(0, 0, d_1)$
5:       $\Delta_1 = U(1, 0, d_2) - U(1, 0, d_1)$
6:       $P(X_{Y=0} = 0) = (\text{Exp. U difference} - \Delta_1)/(\Delta_0 - \Delta_1)$
7:
8:     $d_1, d_2, x', y', q_{\text{crit}} \leftarrow$ Algorithm 1$(U, \Pi_\Sigma^\delta, N, \sigma_1 = \text{do}(Y = 1))$        $\triangleright$ Identify $q_{\text{crit}}$ for do$(Y = 1)$
9:       Exp. U difference $= (U(d_2, x', y') - U(d_1, x', y')) * (1 - 1/q_{\text{crit}})$
10:      $\Delta_0 = U(0, 1, d_2) - U(0, 1, d_1)$
11:      $\Delta_1 = U(1, 1, d_2) - U(1, 1, d_1)$
12:      $P(X_{Y=1} = 0) = (\text{Exp. U difference} - \Delta_1)/(\Delta_0 - \Delta_1)$
13:
14:     $d_1, d_2, x', y', q_{\text{crit}} \leftarrow$ Algorithm 1$(U, \Pi_\Sigma^\delta, N, \sigma_1 = \text{do}(X = 0))$        $\triangleright$ Identify $q_{\text{crit}}$ for do$(X = 0)$
15:      Exp. U difference $= (U(d_2, x', y') - U(d_1, x', y')) * (1 - 1/q_{\text{crit}})$
16:      $\Delta_0 = U(0, 0, d_2) - U(0, 0, d_1)$
17:      $\Delta_1 = U(0, 1, d_2) - U(0, 1, d_1)$
18:      $P(Y_{X=0} = 0) = (\text{Exp. U difference} - \Delta_1)/(\Delta_0 - \Delta_1)$
19:
20:     $d_1, d_2, x', y', q_{\text{crit}} \leftarrow$ Algorithm 1$(U, \Pi_\Sigma^\delta, N, \sigma_1 = \text{do}(X = 1))$        $\triangleright$ Identify $q_{\text{crit}}$ for do$(X = 1)$
21:      Exp. U difference $= (U(d_2, x', y') - U(d_1, x', y')) * (1 - 1/q_{\text{crit}})$
22:      $\Delta_0 = U(1, 0, d_2) - U(1, 0, d_1)$
23:      $\Delta_1 = U(1, 1, d_2) - U(1, 1, d_1)$
24:      $P(Y_{X=1} = 0) = (\text{Exp. U difference} - \Delta_1)/(\Delta_0 - \Delta_1)$
25:
26:     **if** $P(Y_{X=0} = 0) = P(Y_{X=1} = 0)$ **then**        $\triangleright$ Identify $G$ and $P$ from interventionals
27:       **if** $P(X_{Y=0} = 0) = P(X_{Y=1} = 0)$ **then**
28:         $G \leftarrow ()$
29:         $P(x, y) = P(X_{Y=0} = x)P(Y_{X=0} = y)$
30:       **else**
31:         $G \leftarrow (Y \rightarrow X)$
32:         $P(x, y) = P(Y_{X=0} = y)P(X_{Y=y} = x)$
33:       **end if**
34:     **else**
35:       $G \leftarrow (X \rightarrow Y)$
36:       $P(x, y) = P(X_{Y=0} = x)P(Y_{X=x} = y)$
37:     **end if**
38:     **return** $G$, $P(x, y)$
39: **end function**

---

distributions $I$ in the source domain, and ii) (partial) knowledge of the joint $P^*$ and causal graph $G^*$ in the target domain (Pearl & Bareinboim, 2011; Bareinboim & Pearl, 2012b). Hence, these results differ from Theorems 1 and 2 in that they restrict to the case where all assumptions on the data generating process (i.e. inductive biases) can be expressed as (partial) knowledge of the underlying CBN. For example, Bareinboim & Pearl (2016) claim the problem is essentially solved in the case where 'assumptions are expressible in DAG form'. This does not constrain possible approaches to domain generalisation that make use of non-causal assumptions and heuristics[5], and indeed deep learning algorithms exploit a much wider set of inductive biases than causal assumptions alone (Neyshabur et al., 2014; Battaglia et al., 2018; Rahaman et al., 2019; Goyal & Bengio, 2022; Cohen & Welling, 2016). In many real-world tasks these may be sufficient to identify 'good enough' (i.e. regret-bounded) policies without requiring knowledge of the causal structure of the data generating process. Our aim has been to establish if learning causal models is necessary for domain generalisation in general. Hence assuming that agents are restricted to using inductive biases that amount to (partial) knowledge of the underlying CBN would be begging the question.

**Causal hierarchy's theorem (CHT).** The celebrated causal hierarchy theorem (Bareinboim et al., 2022; Ibeling & Icard, 2021) shows that for almost all environments there are causal relations between environment variables that cannot be identified from observational data without additional assumptions. Does this imply that a causal model is necessary for identifying optimal policies?

First, note that the CHT is an insufficiency result, and only implies trivial necessity results. For example, is a causal model necessary for identifying all causal and associative relations between environment variables? Yes, but only because this set of observational and interventional distributions *is* a causal model. Formally, we can identify the underlying causal model (up to latent confounders) by assuming causal faithfulness, which holds for almost all causal models (Meek, 2013). The difference here is that the CHT is concerned with the identifiability of all causal and associative relations between environment variables. This sets a much higher bar than domain generalisation, which focuses on identifying a strict subset of these (regret-bounded policies) (Figure 4).

Secondly, the CHT is concerned with the collapse (or lack thereof) of the causal hierarchy. For example, that observational data is insufficient for identifying all causal queries. We do not restrict agents to having observational training data—in fact, typically we assume that agents have access to both observational and interventional data in the online learning setting that we consider (e.g. agents can intervene to fix the decision node $D$ by assumption).

Finally, we can imagine a refinement of the CHT which states that observational data is insufficient for identifying regret-bounded policies without additional assumptions, bringing it in line with Theorems 1 and 2. If this was implied by the CHT, it would not imply our results unless we restrict to the case where all assumptions as constraints on the causal structure (similar to transportability). Likewise, it is simple to show that Theorem 1 does not imply the CHT. In deriving Theorem 1 we do not restrict to observational distributions (or make any restrictions on the data available to the agent when generating its policy).

---

[5]Indeed, notable examples of causal assumptions that go beyond those expressible in DAG form include restricting the classes of structural equations Mooij et al. (2016) and assuming cause-effect asymmetry (Mitrovic et al., 2018)

