# OpenReview forum: "Robust agents learn causal world models"
_ICLR.cc/2024/Conference — ICLR 2024 oral_

### Official Review · Reviewer_VrNg · 2023-10-29

**Soundness:** 3 good
**Presentation:** 3 good
**Contribution:** 3 good
**Rating:** 8
**Confidence:** 3

**Summary:**

The paper shows that any agent that could effectively "learn" the optimal decision under distribution shifts MUST have learned the (approximate) causal model of the data generating process. The implications of this result on the related research areas such as transfer learning and causal inference have been discussed.

**Strengths:**

1. The problem considered is fundamental.

2. The idea is cute and clean.

**Weaknesses:**

Only necessary condition is proved but not the sufficient condition. It will be stronger to prove something like, if the agent has learned some "approximate" causal relationship, it can efficiently learn the optimal decision under distribution shift.

**Questions:**

How to identify and prove the sufficient condition for learning the causal model for learning the optimal decision making under distribution shift?

---

> ### Author Response · Authors · 2023-11-19
> **Reply to reviewer VrNg**
>
> We thank the reviewer for their encouraging review and helpful comments.
>
> As the reviewer has requested, we have derived a new theorem which proves the sufficiency condition corresponding to the necessity condition proven in the original submission. This new theorem is included in the updated manuscript (Theorem 3).

---

> > ### Comment · Reviewer_VrNg · 2023-11-21
> >
> > This is a nice addition to the paper.

---

### Official Review · Reviewer_vFfA · 2023-10-30

**Soundness:** 3 good
**Presentation:** 3 good
**Contribution:** 3 good
**Rating:** 6
**Confidence:** 4

**Summary:**

This paper proposes some theoretical results about decision making tasks that, if the environment is generated by a Bayesian network, and an agent is able to learn a low regret strategy for all mixture of local interventions on the environment, including hard intervention and randomized experiments on any number of variables, then we can recover the causal structure of the environment from the optimal decision learned by the agent. Therefore, if we want to obtain such an agent, it is necessary to learn the causal structure.

**Strengths:**

1. They propose theoretical results connecting decision making and causal structure learning. As suggested by their results, a robust enough agent should always learn the causal structure.
2. The limitation for learning causal structure can be transferred to limitation of robust decision making by their results.
3. Their result gives an example about inferring causal structure when only one variable is observed under each intervention.

**Weaknesses:**

1. They do not conduct an experiment for justifying their results.
2. Their results can only be applied to a small range of scenarios, where we need to reach small regret for all mixture of local interventions. However, most applicable tasks, such as transfer learning, only consider interventions on a subset of variables.
3. There are some spelling mistakes in their text, and some usage of notations are unclear in their text and proof.

**Questions:**

see  in Weaknesses

---

> ### Author Response · Authors · 2023-11-19
> **reply to reviewer vFfa**
>
> We thank the reviewer for their helpful comments, and have made the following changes to the manuscript in response to their recommendations
>
> 1. Experiments section (appendix F.1, end of appendices). We now empirically validate our results using a simulation example as suggested by the reviewer. This involves converting the proof of theorem 1 into an algorithm for learning the underlying CBN from the agents policy under distributional shifts, and testing it on randomly generated CIDs. We also explore how the accuracy of the learned CBN scales with the agent’s regret bound.
>
> 2. New appendix F with simplified overview of proof.
>
> We agree our results would be stronger and more applicable to current systems if we could extend theorems 1 & 2 to shifts on a small subset of environment variables. Unfortunately, this would be a significant extension of the current proofs and we were unable to do this extension in the time available. We agree this is an important direction for future work.

---

### Official Review · Reviewer_4TT8 · 2023-10-31

**Soundness:** 4 excellent
**Presentation:** 4 excellent
**Contribution:** 4 excellent
**Rating:** 10
**Confidence:** 4

**Summary:**

The paper presents theoretical results showing that any agent that learns well under distributional shifts, must have learned the causal structure of the environment. Here distributional shifts that are most important are shifts of the latent causal variables. That is, if one can generalize across the set of possible changes in these variables, one has learned the causal structure. The result is both deep and intuitive, and has widespread implications. Although theoretical in important senses, e.g. it assumes some unspecified learning method, the result is no less powerful in arguing that to transfer one must learn causal structure.

**Strengths:**

This paper is a gem. The theoretical analysis is simple and clear, the implications are broad and powerful.

**Weaknesses:**

The only weakness, in my opinion, is that the statement of the result in the introduction felt pretty slippery. (See detailed comments below.) All of this was satisfyingly resolved, but I do think the paper would benefit from an effort to sharpen that first section.

Details comments:
- Please define these:
"distributional shifts"
"distributionally shifted environments"
"target domains"
"causal modelling and transfer learning"
- " used to derive out results" typo
- "Our analysis focuses on distributional shifts that involve changes to the causal data generating process, and hence can be modelled as interventions (Schölkopf et al., 2021)" This would have been nice in the intro.
- "This does not assume that all shifts an agent will encounter can be modelled as interventions, but requires that the agent is at least capable of adapting to these shifts." I don't know that I understand this sentence.
- "By cCreftheorem: main,theorem: main approx agents" typo?

**Questions:**

I would like to hear what changes to the introduction might look like.

---

> ### Author Response · Authors · 2023-11-19
> **Reply to reviewer 4TT8**
>
> We want to very much thank the reviewer for their encouraging review and helpful comments. We have updated the introduction to be clearer and in line with the reviewers suggestions. On top of this we have also introduced a simplified overview of the proof and an experiments section (appendix F). To answer the reviewers questions,
>
> 1. "Please define these ...". We have added definitions for these terms in sections with the corresponding titles, and removed the term `causal modelling’. If the reviewer would prefer definitions in the form e.g. Definition 1 (Bayesian networks) we will include these.
> 2. Have clarified in the introduction that we are focusing on interventional shifts.
> 3. "This does not assume that all shifts... I don't know that I understand this sentence". Apologies, this was confusingly written. What we are trying to say is, imagine an agent that is robust to `all’ distributional shifts (including those that cannot be represented with local interventions, such as changing the set of variables). This agent will also be subject to our theorems, because it is robust to all distributional shifts which includes interventional shifts as a subset. So our results apply to agents that are robust to a larger set of shifts. Will remove if the reviewer recommends.
> 4. Typos fixed

---

> > ### Comment · Reviewer_4TT8 · 2023-11-22
> > **Thanks for the detailed response**
> >
> > Thank you for the detailed response. The revisions look great. Very cool work.
> >
> > FYI - At the beginning of section 5 there appears to be a note that I don't think you intended to include.

---

### Official Review · Reviewer_gS4b · 2023-10-31

**Soundness:** 2 fair
**Presentation:** 3 good
**Contribution:** 3 good
**Rating:** 8
**Confidence:** 3

**Summary:**

This paper shows a formal connection between generalizing under distribution shits and learning causal models, a connection that has been expressed as hypothesis before (e.g. in Schölkopf 2021). Specifically, they show that if the agent performs well under distribution shifts (bounded regret), then it must have learned a representation that captures the causal structure of the world - in this case, the conditional independencies and causal relationships in the true causal bayesian network.

**Strengths:**

This paper makes an original and significant theoretical contribution by formally establishing a fundamental connection between causal learning and generalisation under distribution shifts.
## Originality:
* They provide a proof for showing that an agent that is sufficiently adaptive has learned a causal model of the environment. This is an impressive achievement and a stronger statement than the one stated by good regulator theorem (which as the authors have cited, has been misunderstood and misrepresented in the past)
## Quality:
* The theoretical results are technically strong, with detailed proofs provided in the appendices.
* The assumptions are clearly stated and well-motivated. The analysis meaningfully relaxes the assumption of optimality.
* The writing is clear, well-structured, and accessible given the technical nature of the work.
## Clarity:
* The paper is well written and easy to read.
## Significance:
* The results have important implications in safety and robustness under distribution shifts.
* The proof is non-trivial and provides a great stepping stone for extending to richer settings (e.g. mediated decision tasks)

**Weaknesses:**

- As the authors acknowledge, the results are mainly theoretical. Even a minimal empirical validation of the key insights would strengthen the paper. For example it would be great even if you turn the informal overview (appendix C) into a simple simulation example rather than remain a thought experiment.
- The scope is currently limited to unmediated decision tasks. Extending the results to broader RL settings would increase applicability (although I acknowledge that seems significantly more challenging task and out of scope of this work - it’s just a personal curiosity at this point and would be excited to see the next paper already).
- The proof is still quite challenging to understand and I believe that there a more informal / simplified sketch that can be introduced to help the reader before dive into the more formal proof.
- On a similar note, the implications of the assumptions are not discussed. (e.g. I’d like to see things like, “assumption 2 implies that there exist distribution shifts for which the optimal policies are different”.

**Questions:**

(apologies for repetition from weaknesses)
- The implications of the assumptions are not discussed. (e.g. I’d like to see things like, “assumption 2 implies that there exist distribution shifts for which the optimal policies are different”.
- "The environment is described by a set of random variables C..." this sentence belongs to the main text since it you don't explain C random variable although it's heavily used.
- Although discussed in the appendix, i'd like to see the description of what squares, circles and diamonds mean in the CID
- In assumption 1 you stated $\text{Desc}_D \cap \text{Anc}_U = \emptyset$ but this doesn't exclude the trivial setting (which you state in the appendix is not of focus). Can you either extend the assumption or comment in the main text that it's not of interest the trivial setting? (i know it's a nitpick but got me wondered while reading it in the main text and i feel that since you thought about it you could have mentioned it earlier in the text).
- Definition 6 in the appendix: Shouldn't it be $\mathbb{E}^{\pi_\sigma}$ (subscript on policy)? Also, $\delta \geq 0$ is missing.
- Can you please give a simple sketch of the proof? This would help the readability significantly. Also i feel there is a simple sentence that can be written on each theorem that explains its implications

---

> ### Author Response · Authors · 2023-11-19
> **Reply to reviewer gS4b**
>
> We thank the reviewer for their thoughtful review, and have made the following changes to the manuscript in response to their recommendations
>
> 1. Experiments section (appendix F.1, end of appendices). We now empirically validate our results using a simulation example as suggested by the reviewer. This involves converting the proof of theorem 1 into an algorithm for learning the underlying CBN from the agents policy under distributional shifts, and testing it on randomly generated CIDs. We also explore how the accuracy of the learned CBN scales with the agent’s regret bound.
>
> 2. Simplified sketch of proof (appendix F, end of appendices). The previous proof overview was too involved, as we have re-written it to be 3 paragraphs long. The example covers a simple binary decision task with two latent variables. There is also an algorithm in appendix F.1 detailing how to learn the CBN in this setting.
>
> 3. Discussed implications of assumptions immediately after their statement and suggested.
>
> 4. Added missing description of chance nodes “C” on page 3.
>
> 5. Added description of CID nodes (circles, squares diamonds) to caption of figure 1.
>
> 6. Added sentence description immediately after / before each theorem.
>
> 7. The reviewer also points out that it was not made clear how our assumptions deal with the trivial case (i.e. where the agent’s action does not cause the utility). In this case, all policies are optimal, and hence the assumption of domain independence is violated (there is a single policy that is optimal under all domain shifts). We have added a note making this clear.
>
> 8. Typos corrected
>
> We tried to extend the results to unmediated decision tasks, but unfortunately this was too challenging in the time available. We agree this is an important direction for future work.

---

> ### Author Response · Authors · 2023-11-21
>
> As the discussion period comes to an end, we would greatly appreciate any response to the changes made, which include the requested experiments and proof overview (summarised here and presented in appendix F). Either way, many thanks for the helpful review.

---

### Author Response · Authors · 2023-11-19
**Global response to reviewers and AC**

We thank the reviewers for their invaluable feedback, and are encouraged that they describe the paper as "an impressive achievement", " an original and significant theoretical contribution" that has "important implications in safety and robustness" (Reviewer gS4b), and that "The problem considered is fundamental. The idea is cute and clean" (Reviewer VrNg), and "This paper is a gem. The theoretical analysis is simple and clear, the implications are broad and powerful." (Reviewer 4TT8).

**main changes to updated manuscript**
(details and additional changes in individual replies)

1. Empirical evaluation of results, learning the underlying CBN from agent's policy for randomly generated CIDs (Appendix F.1). Application to causal discovery including pseudocode for Theorem 1. Exploration of how the regret bound effects the accuracy of the learned model.
2. Simplified and shortened overview of proof (Appendix F).
3. Extended sufficiency theorem to cover approximate causal models (Theorem 3).
4. Updated introduction, interpretation and discussion sections. Updated figures.

---

### Meta-Review · Area_Chair_wyN4 · 2023-12-06

**Metareview:**

The paper elucidates a widely conjectured connection between causality and robustness in the area of decision making agents. Specifically, the authors show that "any agent capable of satisfying a regret bound under a large set of distributional shifts must have learned an approximate causal model of the data generating process, which converges to the true causal model for optimal agents."

This is a strong and interesting results which has unanimously recognized by the reviewers.

**Justification For Why Not Higher Score:**

NA

**Justification For Why Not Lower Score:**

The reviewers were all on the accept side, some with high or maximal scores, which are well justified given the relevancy of the results. This paper should be given a special stage at ICLR.

---

### Decision · Program_Chairs · 2024-01-16

Accept (oral)